# Recurrent Inertial Graph-Based Estimator (RING): A Single Pluripotent Inertial Motion Tracking Solution

**Simon Bachhuber**                                                   *simon.bachhuber@fau.de*
*Department of Artificial Intelligence in Biomedical Engineering*
*Friedrich-Alexander-Universität Erlangen-Nürnberg*

**Ive Weygers**                                                       *ive.weygers@fau.de*
*Department of Artificial Intelligence in Biomedical Engineering*
*Friedrich-Alexander-Universität Erlangen-Nürnberg*

**Dustin Lehmann**                                                   *dustin.lehmann@tu-berlin.de*
*Control Systems Group*
*Technical University Berlin*

**Mischa Dombrowski**                                                *mischa.dombrowski@fau.de*
*Department of Artificial Intelligence in Biomedical Engineering*
*Friedrich-Alexander-Universität Erlangen-Nürnberg*

**Thomas Seel**                                                      *thomas.seel@imes.uni-hannover.de*
*Institute of Mechatronics Systems*
*Leibniz Universität Hannover*

**Reviewed on OpenReview:** *https://openreview.net/forum?id=h2C3rkn0zR*

## Abstract

This paper introduces a novel ML-based method for Inertial Motion Tracking (IMT) that fundamentally changes the way this technology is used. The proposed method, named RING[1] (Recurrent Inertial Graph-Based Estimator), provides a pluripotent, problem-unspecific plug-and-play IMT solution that, in contrast to conventional IMT solutions, eliminates the need for expert knowledge to identify, select, and parameterize the appropriate method. RING's pluripotency is enabled by a novel online-capable neural network architecture that uses a decentralized network of message-passing, parameter-sharing recurrent neural networks, which map local IMU measurements and nearest-neighbour messages to local orientations. This architecture enables RING to address a broad range of IMT problems that vary greatly in aspects such as the number of attached sensors, or the number of segments in the kinematic chain, and even generalize to previously unsolved IMT problems, including the challenging combination of magnetometer-free and sparse sensing with unknown sensor-to-segment parameters. Remarkably, RING is trained solely on simulated data, yet evaluated on experimental data, which indicates its exceptional ability to zero-shot generalize from simulation to experiment, while outperforming several state-of-the-art problem-specific solutions. For example, RING can, for the first time, accurately track a four-segment kinematic chain (which requires estimating four orientations) using only two magnetometer-free inertial measurement units. This research not only makes IMT more powerful and less restrictive in established domains ranging from biomechanics to autonomous systems, but also opens its application to new users and fields previously untapped by motion tracking technology. Code and data is available here.

---

[1]one to track them all

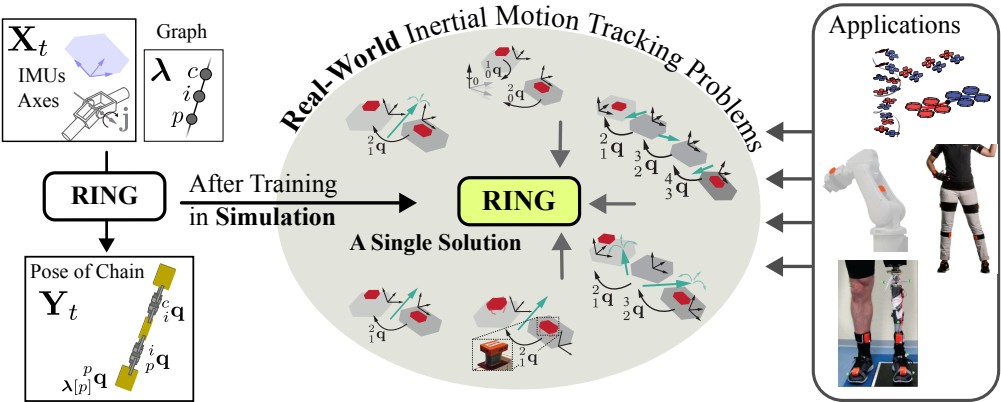

Figure 1: RING is a ML-based method that provides a versatile, pluripotent IMT solution applicable across a broad range of challenging IMT problems, designed for use without the need for expert knowledge. Remarkably, RING is trained solely on simulated data, yet zero-shot generalizes to real-world experiments and outperforms several problem-specific state-of-the-art solutions.

## 1 Introduction

In the domain of multi-agent systems, structural policies have shown great potential for the control of complex agents; meanwhile, Recurrent Neural Networks (RNNs) are an established choice for sequential data. The potential of their combination for the analysis of structural sequential data has rarely been investigated and exploited. This combination seems particularly promising for state estimation in graph-structured systems, such as, for example, for IMT of Kinematic Chains (KCs).

The need for reliable and accurate estimation of the orientation, attitude, or pose of articulated objects in three-dimensional (3D) space spans across various application domains ranging from aerospace engineering (Euston et al., 2008; Givens & Coopmans, 2019) to health applications (Buke et al., 2015; López-Nava & Muñoz-Meléndez, 2016; Seel et al., 2020). Inertial Measurement Units (IMUs), which typically comprise a 3D accelerometer, a 3D gyroscope, and a 3D magnetometer, have become smaller and less expensive within the last two decades and have therefore rapidly become the most promising technology for accurate, reliable, and inexpensive motion tracking in rigid bodies and KCs, especially since camera-based systems are typically more expensive, more restrictive, and suffer from occlusion (von Marcard et al., 2017; Huang et al., 2018).

However, fusing the available measurement signals to estimate the desired orientations requires advanced IMT algorithms that typically need to overcome a combination or all of the following three main IMT challenges (Seel et al., 2020): (1) inhomogeneous magnetic fields in indoor environments and in proximity of ferromagnetic material or electric devices; (2) sensor-to-segment calibration, i.e. identifying the joint position and axis orientations in local sensor coordinates; (3) solving sparse problems in which some segments of the KC are not equipped with a sensor, to improve the usability and reduce costs.

In recent years, numerous highly specialized methods have been proposed to address these challenges. Magnetometer-free methods have been developed to estimate the relative orientation between two adjacent segments by exploiting different kinematic constraints (Kok et al., 2014; Laidig et al., 2017; Lehmann et al., 2020; 2024). Moreover, numerous general-purpose magnetometer-free attitude estimators have been proposed (Mahony et al., 2008; Madgwick, 2010; Seel & Ruppin, 2017; Weber et al., 2021; Laidig & Seel, 2023). Several algorithms were developed to achieve sensor-to-segment calibration for specific kinematics with full sensor setups (Taetz et al., 2016; McGrath et al., 2018; Olsson et al., 2020). Finally, a variety of sparse IMT methods have been developed that either use a limited number of sensors while still depending on magnetometers (von Marcard et al., 2017; Huang et al., 2018; Sy et al., 2020; 2021; Zheng et al., 2021) or are magnetometer-free (Grapentin et al., 2020; Yi et al., 2021; 2022; Bachhuber et al., 2023; Van Wouwe et al., 2023).

In summary, there exists a plethora of methods, each tailored to a very specific application, such as magnetometer-free tracking of a single-degree-of-freedom (1-DoF) joint (Lehmann et al., 2020), or human

pose estimation from six IMUs, as detailed in Yi et al. (2021). To apply IMT, the user must successfully identify the method that is suitable for the given problem and typically specify various parameters such as joint axes directions and sensor placement. Therefore, the user must be an expert in the field of IMT, which strongly limits the use of IMUs in many application domains. To make matters even worse, a given problem might require a nontrivial combination of methods which may exclude each other, e.g., the tracking of a sparse three-segment KC currently requires known joint axes directions (Bachhuber et al., 2023), but the method that estimates the joint axes directions does not allow for a sparse sensor setup (Olsson et al., 2020).

What if, instead of a plethora of methods, we had a single *pluripotent* method that can be used for, e.g., magnetometer-free tracking of 1-DoF joints, and the tracking of sparse sensor setups with known or even unknown joint axes directions? What if we had *one to track them all*?

In this work, we demonstrate that ML methods can be used to, for the first time, achieve this goal. We propose a method, named RING, that combines a novel neural network architecture, named RINGCell, with an elaborate training data simulation, the Random Chain Motion Generator (RCMG). The key insight that enables the pluripotency of RING is that a given system in an Inertial Motion Tracking Problem (IMTP) can be viewed as a graph where nodes represent segments and edges represent a single DoF. Then, a shared set of parameters can be applied on a decentralized, per-node level, where only local IMU measurements are observed and only the local estimation problem is solved, i.e., the orientation relative to the parent is estimated. Information exchange between nodes is enabled by passing messages along the edges of the graph.

We show that RING can be used to plug-and-play solve a range of challenging magnetometer-free sparse or non-sparse motion tracking problems with a single trained neural network and that it even solves previously unsolved challenging IMTPs, such as, e.g., the tracking of a triple-hinge-joint system with only two IMUs. We demonstrate that although RING is trained solely on simulated data, it zero-shot generalizes to experimental data and aligns with various state-of-the-art (SOTA) results.

## 2 Related Works

As mentioned above, there exists a plethora of highly specialized methods in the field of IMT, but there is no single pluripotent solution that solves a variety of IMTPs. Even the use of ML methods for IMT has so far only led to specific solutions for single IMTPs. RNNs have been used in Weber et al. (2021) to achieve SOTA attitude estimation, and in Bachhuber et al. (2023) to successfully track a specific sparse KC. Deep learning has also been used for human motion capture, where the full-body pose is estimated from typically six or more IMUs, and previous work has shown promising results (von Marcard et al., 2017; Huang et al., 2018; Zheng et al., 2021; Yi et al., 2021; 2022; Van Wouwe et al., 2023; Puchert & Ropinski, 2023). However, while addressing a challenging problem, these methods are limited to human motion capture with one specific sensor setup and assume statistical patterns of human motion (von Marcard et al., 2017), or full-body biomechanical models (Yi et al., 2022) to constrain the estimated pose.

From a methodological viewpoint, RING uses a decentralized network of message-passing RNNs with shared parameters that are trained via supervised learning. The concept of decentralized networks, communication, and collaboration is at the heart of multi-agent systems, and the means of communication can either be prescribed (Panait & Luke, 2005; Wang et al., 2019) or, more recently, learned (Sukhbaatar et al., 2016; Foerster et al., 2016; Wang et al., 2018; Pathak et al., 2019; Huang et al., 2020). In deep Reinforcement Learning (RL), feedforward networks have been used to parameterize structured policies that pass messages along the edges of a graph in Sukhbaatar et al. (2016); Foerster et al. (2016); Wang et al. (2018); Huang et al. (2020), and distinct advantages of message-passing have been shown. In particular, in Huang et al. (2020) it was investigated whether centralised control can emerge from decentralized policies, and they show that it is possible to learn a global policy that achieves locomotion across various agent morphologies. It is interesting to note that policies must collaborate in order to achieve a global task, e.g., locomotion, and are motivated by a global reward. In the present work, the decentralized RNNs must collaborate by exchanging information in order to solve the task of motion tracking, and are motivated by a decentralized loss function. The advantages of communication and global coordination emerging in a decentralized structure were also investigated in Sukhbaatar et al. (2016); Foerster et al. (2016). In particular, the work of Sukhbaatar et al. (2016) uses supervised learning instead of RL for learning communication protocols.

## 3   Preliminaries

### 3.1   Notation

We use a typical notation with scalars denoted by $x$, vectors by $\boldsymbol{x}$, matrices (or higher dimensional tensors) by $\mathbf{X}$, and quaternions (or higher dimensional arrays of quaternions) by $\mathbf{q}$. Additionally, note that the symbol $\mathbb{1}$ defines either the unity element of a given space, or the indicator function, such that, e.g., $\mathbb{1}_0(i)$ is one for $i = 0$ and zero else. The symbol $\otimes$ is used to denote the direct product of vector spaces, and additionally denotes quaternion multiplication, see Definition B.2. Further details can be found in Appendix A.1.

### 3.2   An Inertial Motion Tracking Problem

We define an IMTP as the task of estimating the trajectory of the complete rotational state of an Articulated Rigid-Body System (ARBS) from inertial data. Conceptually, an ARBS is a collection of multiple rigid (or assumed to be rigid) bodies that are interconnected via joints that allow for relative motion between these bodies. Let there be a singleton, inertial reference coordinate system named *base* (sometimes *world*, or *Earth*), then an orientation of a body relative to the base is referred to as an absolute orientation. An orientation of a body relative to the coordinate system of another body is referred to as a relative orientation. Then, assuming that no additional information about the types of joints is provided, to fully describe the rotational state of an ARBS that consists of $N$ bodies at a single moment in time, $N$ orientations must be specified. In this work, we will utilize $N-1$ relative orientations and one absolute orientation.

An IMTP is solved by estimating the rotational state from at most $N$ IMUs that are connected to the bodies of the ARBS (at most one IMU per body), and that provide 3D measurements of angular rates, specific forces, and the magnetic field density in their local sensor coordinates. A magnetometer-free method solves an IMTP without the use of the magnetometer, and such IMUs are referred to as 6D IMUs. In this case, the rotational state of the ARBS can only be estimated up to an absolute heading error, that is, up to an arbitrary rotation around the gravity (or vertical) direction, and the one absolute orientation is referred to as the absolute attitude. This is because both accelerometer and gyroscope measurements are invariant under a rotation around the vertical direction. Finally, note that if at least one body of the ARBS does not have an IMU attached, then the IMTP is said to be sparse.

### 3.3   Graph Connectivity

The topology of an ARBS is represented by a Connectivity Graph (CG) (Featherstone, 2008; 2010), which is an undirected graph where the nodes represent the bodies that constitute the ARBS and the edges represent its joints.

Before the CG can be encoded, the bodies must be numbered. We adopt the broadly adopted standard numbering scheme and notation from Featherstone (2008; 2010) which for an ARBS with $N$ bodies proceeds as follows:

1. The base is assigned the number 0 and it serves as the root node.

2. The remaining bodies are consecutively numbered from 1 to $N$, so that each body has a higher number than its parent.

The CG may then be encoded via a parent array $\boldsymbol{\lambda} \in \mathbb{N}^N$ where $\boldsymbol{\lambda}[i]$ is the body number of the parent of body $i$. Additionally, we define the function $\mu(i)$ to return the set of the body numbers of the children of body $i$, that is

$$\mu_\lambda(i) = \{j \mid \boldsymbol{\lambda}[j] = i \quad \forall j = 1 \dots N\}. \tag{1}$$

**Definition 3.1.** The body $i$ of an ARBS with parent array $\boldsymbol{\lambda}$ is said to be an outer body if it has no children bodies, i.e., $\mu_\lambda(i) = \{\}$, or if its parent is the base, i.e., $\boldsymbol{\lambda}[i] = 0$; otherwise it is said to be an inner body.

As an example, consider an ARBS that is a three-segment KC (see Figure 2). There, the three bodies are numbered increasingly from top-to-bottom, and then, for this numbering, the parent is given by $\boldsymbol{\lambda} = (0, 1, 2)^\intercal$.

Note that if the inner body (middle segment) is assumed to connect to the base, then the parent array is different and given by $\boldsymbol{\lambda} = (0, 1, 1)^\intercal$.

### 3.4 Minimal and Maximal Coordinates

We refer to the generalized coordinates position vector as the minimal coordinates, denoted by $\boldsymbol{q}$. It fully captures the kinematic state of the ARBS using a minimal amount of coordinates. The size of $\boldsymbol{q}$ depends on the DoFs of the joints in the ARBS. In this work, we will consider ARBSs that can move freely in space (without any constraints). Therefore, in the corresponding CG, the edges that connect to the base are 6-DoF free joints and all the remaining edges represent single DoF.

**Definition 3.2.** For an ARBS with $N$ bodies with a CG given by $\boldsymbol{\lambda}$ where the joints that connect to the base are 6-DoF and 1-DoF otherwise, the size of $\boldsymbol{q}$ is given by $N_q = \sum_i^N 7\mathbb{1}_0(\boldsymbol{\lambda}[i]) + (1 - \mathbb{1}_0(\boldsymbol{\lambda}[i]))$. Similarly, the size of the velocity of minimal coordinates $\dot{\boldsymbol{q}}$ is given by $N_{\dot{q}} = \sum_i^N 6\mathbb{1}_0(\boldsymbol{\lambda}[i]) + (1 - \mathbb{1}_0(\boldsymbol{\lambda}[i]))$.

For an ARBS with $N$ bodies, we refer to the set of all euclidean positions and rotational states from the base to all bodies in the system as the maximal coordinates, denoted by $\boldsymbol{t} \in \left(\mathbb{H} \otimes \mathbb{R}^3\right)^N$. The size of $\boldsymbol{t}$ depends only on the number of bodies in the system.

### 3.5 Representing Orientations

We represent 3D orientations with quaternions $\mathbf{q}$ due to various advantages over equivalent representations using orthogonal matrices or Euler angles (Kuipers, 2002). In particular, we use ${}^0_1\mathbf{q} \in \mathbb{H}$ to denote the absolute orientation from the base to the body one's coordinate system. Similarly, we use ${}^1_2\mathbf{q}$ to denote the relative orientation from body one to body two. In Appendix B, we define various utilized quaternion-related operations.

### 3.6 Loss Function For Orientations

In order to train and evaluate ML methods that predict orientations (represented by quaternions), a suitable metric function must be identified. We can compare the difference between a ground truth orientation $\mathbf{q}$ and the corresponding predicted orientation $\hat{\mathbf{q}}$ by computing the angle of the smallest rotation that makes ground truth and prediction identical. It is given by $\texttt{angle}(\mathbf{q} \otimes \hat{\mathbf{q}}^*)$ where $\otimes$ denotes quaternion multiplication, $*$ denotes the complex conjugate, and $\texttt{angle}$ extracts the rotation angle of a quaternion (see Appendix B.5). Then, we use the following mean-squared-error function $\texttt{loss} : \mathbf{q} \in \mathbb{H}, \hat{\mathbf{q}} \in \mathbb{H} \to \mathbb{R}_{\geq 0}$ to calculate a scalar error between two quaternions, given by

$$\texttt{loss}(\mathbf{q}, \hat{\mathbf{q}}) = \texttt{angle}(\mathbf{q} \otimes \hat{\mathbf{q}}^*)^2. \tag{2}$$

The loss function for a single orientation in eq. (2) is then used to compute the mean-squared-error for the entire rotational state of the KC in eq. (10).

## 4 Method

In this section, we define the problem under consideration and the proposed method which consists of three components:

- A virtually infinite, simulated training data set (see Section 4.2).

- A novel, online-capable neural network architecture, named RINGCell, purpose-built for state estimation in ARBS (see Section 4.3). In contrast to typical RNNs that map a fixed number of input features to a fixed number of output features using a centralized logic, RINGCell leverages the graph-structure of ARBS and employs a decentralized, message-passing logic with a shared set of parameters. This design maps a fixed number of input features *per body* to a fixed number of output features *per body*, and it allows RINGCell to adapt to the size and topology of the ARBS without the need for retraining.

```
sys = {
    n, λ ∈ ℕ^N,
    J ∈ ℝ^{N×3},
    R ∈ ℝ^{2N×3},
    K, Γ ∈ ℝ^{N×6},
    T_s ∈ ℝ,
}
```

Figure 2: System object example (see Definition 4.2) for a three-segment KC with $N = 3$ bodies and parent array $\boldsymbol{\lambda}_3 = (0, 1, 2)^\intercal$.

- A powerful IMT solution, named RING, that can solve a broad range of IMTPs in the real world (see Section 4.4). RING is obtained by training RINGCell on the simulated training data.

Finally, Section 4.5 provides a software implementation of the entire method and, more specifically, RING. Most notably, we provide a single file that trains the here benchmarked version of RING from scratch *without requiring any files that contain real-world training data.*

## 4.1 Problem Formulation: A Class of Inertial Motion Tracking Problems

In this work, we consider the following class of IMTPs. We consider an $N$-segment KC where $N \in \{1, 2, 3, 4\}$ with unknown physical geometry and with segments that are interconnected via hinge joints. The axes directions of these hinge joints may be *known or unknown.* For each inner body *at most one* and for each outer body exactly one 6D IMU, with unknown sensor-to-body position, is *rigidly or nonrigidly* attached to each body. For each body and with the KC at rest, the constant sensor-to-body orientation is assumed to be known. The initial pose of the ARBS is assumed to be unknown. Then, the task is to estimate, for every timestep, the rotational state of the KC (consisting of $N$ orientations) from the available IMU measurements and the available joint axes directions. Note that the task requires providing accurate estimates at each timestep, thus necessitating an online-capable solution focused on online, real-time processing (filtering) as opposed to retrospective data processing (smoothing). To summarize, two IMTPs of this class of IMTPs can differ in: 1) the number of bodies $N$ in the KC, 2) the number of attached IMUs, 3) the number of known joint axes directions, and 4) whether IMUs are attached rigidly or nonrigidly. A broad range of IMTPs from this class of IMTPs are illustrated in Figure 1 within the grey ellipsoid.

**Definition 4.1.** The parent arrays $\boldsymbol{\lambda}_N$ of the $N$-segment KC where $N \in \{1, 2, 3, 4\}$ are given by, without loss of generality, $\boldsymbol{\lambda}_1 = (0)^\intercal, \boldsymbol{\lambda}_2 = (0, 1)^\intercal, \boldsymbol{\lambda}_3 = (0, 1, 2)^\intercal, \boldsymbol{\lambda}_4 = (0, 1, 2, 3)^\intercal$, respectively.

## 4.2 Training Data: The RCMG Algorithm

The RING network is trained on data obtained from simulated random motion of one-, two-, three-, and four-segment KCs. The procedure that generates this training data (from only PseudoRNG) is called the Random Chain Motion Generator (RCMG) (Bachhuber et al., 2022). The RCMG procedure (see Algorithm 1) has three main steps that execute consecutively:

1. Randomized KC (see Section 4.2.1). A randomized KC is drawn to manipulate the downstream simulation and achieve various forms of domain randomization that enable to bridge the sim-to-real gap from simulation (training) to real-world (testing).

2. Random Motion Generation (see Section 4.2.2). The KC is simulated to randomly move in space and the maximal coordinates of all bodies relative to the base are computed for every timestep.

3. Get $\mathbf{X}, \mathbf{Y}$ Data (see Section 4.2.3). From the maximal coordinates of all bodies, the IMU, joint axes, and pose data is computed, and returned as training data.

Internally, the RCMG procedure uses a system object `sys` (see Definition 4.2) which is the collection of various attributes such as, e.g., a joint axes array $\mathbf{J} \in \mathbb{R}^{N \times 3}$.

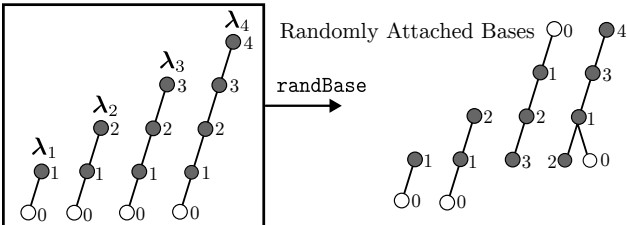

Figure 3: Each sequence of training data is simulated using the RCMG. In the first step of the RCMG a randomized KC is created which involves randomizing the node in the KC to which the base attaches. Afterwards, the nodes in the graph are numbered according to Section 3.3. For example here, for $\boldsymbol{\lambda}_4$, the new numbering array is $\boldsymbol{n} = (2, 1, 3, 4)^\intercal$, and the new parent array $\boldsymbol{\lambda} = (0, 1, 1, 3)^\intercal$.

**Definition 4.2.** We define a system object `sys` with $N$ bodies as the collection of the following attributes:

- a numbering array of integers $\boldsymbol{n} \in \mathbb{N}^N$ which stores a permutation of the numbers going from $1 \ldots N$,

- a parent array of integers $\boldsymbol{\lambda} \in \mathbb{N}^N$,

- a joint axes array $\mathbf{J} \in \mathbb{R}^{N \times 3}$ that contains the hinge joint axis direction,

- a relative-to-parent position array $\mathbf{R} \in \mathbb{R}^{2N \times 3}$ that contains the position vector of the body's coordinate system relative to its parent (expressed in the parent's coordinate system). In Section 4.2.1, it is outlined that for each of the $N$ bodies there is a second IMU body. In the $\mathbf{R}$ array, the first $N$ values are used to specify the position vector between two non-IMU bodies (segment-to-segment positions, physical geometry of the KC), while the last $N$ values are used to specify the position vector from non-IMU body to the corresponding IMU body (segment-to-body positions, IMU attachment). This is done to independently randomize both the physical geometry and the IMU attachment, forcing the network to learn to generalize to scenarios such as an arm robot with unknown segment lengths, as well as to calibrate for an unknown IMU attachment,

- an array of stiffness parameters for $N$ free joints $\mathbf{K} \in \mathbb{R}^{N \times 6}$,

- an array of damping parameters for $N$ free joints $\boldsymbol{\Gamma} \in \mathbb{R}^{N \times 6}$,

- a float representing the sampling time $T_s \in \mathbb{R}$, here always $0.01 \, \text{s}$.

An exemplary system object is shown in Figure 2.

### 4.2.1 Step 1) Randomized Kinematic Chain

In order to enrich the simulated training data, several forms of domain randomization are achieved by drawing a KC with randomized system attributes.

The first domain randomization is the randomization of all downstream forward kinematics applications, and additionally, randomization of the absolute random translation and orientation in the generation of random motion. This has been shown to improve the training data such that the trained network more effectively closes the sim-to-real gap (Bachhuber et al., 2023) This is achieved by re-attaching the bases randomly and afterwards, the nodes in the graph are re-numbered according to Section 3.3. An example of this is shown in Figure 3. Secondly, the $N$ randomized hinge joint axes $\mathbf{J}$ are drawn. Thirdly, the position array $\mathbf{R}$ is randomized by drawing values from uniform ranges.

Finally, the stiffness $\mathbf{K}$ and damping $\boldsymbol{\Gamma}$ arrays are randomized. These values are used to model nonrigidly attached IMUs by connecting additional nodes that are connected via passive spring-damper free joints. For each node $i$ in the CG, we add an additional IMU node with body number $i + N$ that is a child of node $i$ and that is connected to node $i$ via a passive spring-damper free joint. The stiffness $\mathbf{K}[i]$ and damping $\boldsymbol{\Gamma}[i]$

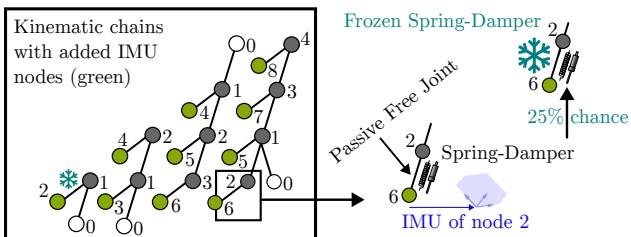

Figure 4: For each node $i$ in the KCs, there exists a second IMU node (that is not counted in the body count $N$ of the system; here in green) with body number $i + N$. The IMU node is connected via a passive spring-damper free joint to the original node in order to simulate nonrigidly-attached IMUs. There is a 25% chance that the damping and stiffness parameters of the passive free joint are chosen such that the IMU is effectively rigidly attached and the passive free joint is frozen.

parameters are used during the forward simulation of the KC as the parameters of the 3D spring-damper dynamics between node $i$ and (IMU) node $i + N$. For each node $i$, the stiffness $\mathbf{K}[i]$ and damping $\mathbf{\Gamma}[i]$ parameters are randomized in a way such that, either the IMU is effectively rigidly attached, or such that the IMU moves relative to the segment node $i$ (which then models nonrigid attachment).

### 4.2.2   Step 2) Random Motion Generation

As a second step, the RCMG procedure generates random motion of the previously obtained randomized KC. Random motion is obtained by drawing a random reference trajectory for all joints in the KC. The generation of such randomized reference trajectories is influenced and constrained by various parameters, e.g., upper limits on angular velocities or lower limits on the amount of motion (to avoid jittering). Additional details on the reference trajectory generation can be found in Appendix C.1. Afterwards, a dynamical forward simulation is performed where joint forces are computed using PD control such that the random reference is tracked. Note that the additionally added nodes to model nonrigid IMU attachment are passive free joints and as such they are not actuated. Finally, the trajectories of maximal coordinates of all $N$ bodies and $N$ IMU bodies, given by $\mathbf{T} \in \left( \mathbb{H} \otimes \mathbb{R}^3 \right)^{2N \times T}$ (from base to body), are computed.

### 4.2.3   Step 3) Get $\mathbf{X}, \mathbf{Y}$ Data

In the last step, the training tuples of $\mathbf{X}, \mathbf{Y}$ are computed from the trajectories of maximal coordinates and afterwards returned. They are:

- $\mathbf{X} \in \mathbb{R}^{T \times N \times 9}$, where $\mathbf{X}[:, i, :6]$ is the simulated 6D IMU data for body $i$ as measured in its IMU body $i + N$ (but for each inner body, there is a $\frac{2}{3}$ chance that the IMU data gets dropped out and replaced by zeros), and where $\mathbf{X}[:, i, 6:]$ is the joint axis direction $\mathbf{J}[i]$ of the hinge joint between body $i$ and its parent $\mathbf{\lambda}[i]$ and zeros if the parent is the base (but for each body, there is a $\frac{1}{2}$ chance that the joint axis direction data gets dropped out and replaced by zeros), and

- $\mathbf{Y} \in \mathbb{H}^{T \times N}$ where $\mathbf{Y}[:, i]$ is the timeseries of: 1) absolute attitudes ${}^i_0\mathbf{q}(t)$ if the parent $\mathbf{\lambda}[i]$ is the base; 2) relative orientations ${}^i_p\mathbf{q}(t)$ from body $i$ to its parent $p = \mathbf{\lambda}[i]$, and

where $T$ is the number of timesteps (here, $60\,\mathrm{s}$ at $100\,\mathrm{Hz}$, thus $T = 6000$), and $N$ is the number of bodies in the KC. IMU and joint axes data is dropped out in order to force RING to learn to solve IMTPs with sparse IMU placement (an inner body may not have an IMU attached to it), and learn to solve IMTPs with require sensor-to-segment calibration. Finally, multiple sequences are stacked to build a training batch. These input and output arrays ($\mathbf{X}$ and $\mathbf{Y}$) directly align with the provided software implementation of RING, see Appendix 4.5.

---

**Algorithm 1** `RCMG` (Generate One Training Data Pair)

---
1: **Input:** $\boldsymbol{\lambda}_N$
2: **Output:** $\mathbf{X} \in \mathbb{R}^{T \times N \times 9}$, $\mathbf{Y} \in \mathbb{H}^{T \times N}$
3: sys $\leftarrow$ `randSys`($\boldsymbol{\lambda}_N$) {see Algorithm 2}
4: $\mathbf{T} \leftarrow$ `randMotion`(sys) {see Algorithm 3}
5: $\mathbf{X}, \mathbf{Y} \leftarrow$ `getXY`(sys, $\boldsymbol{\lambda}_N$, $\mathbf{T}$) {see Algorithm 4}

---

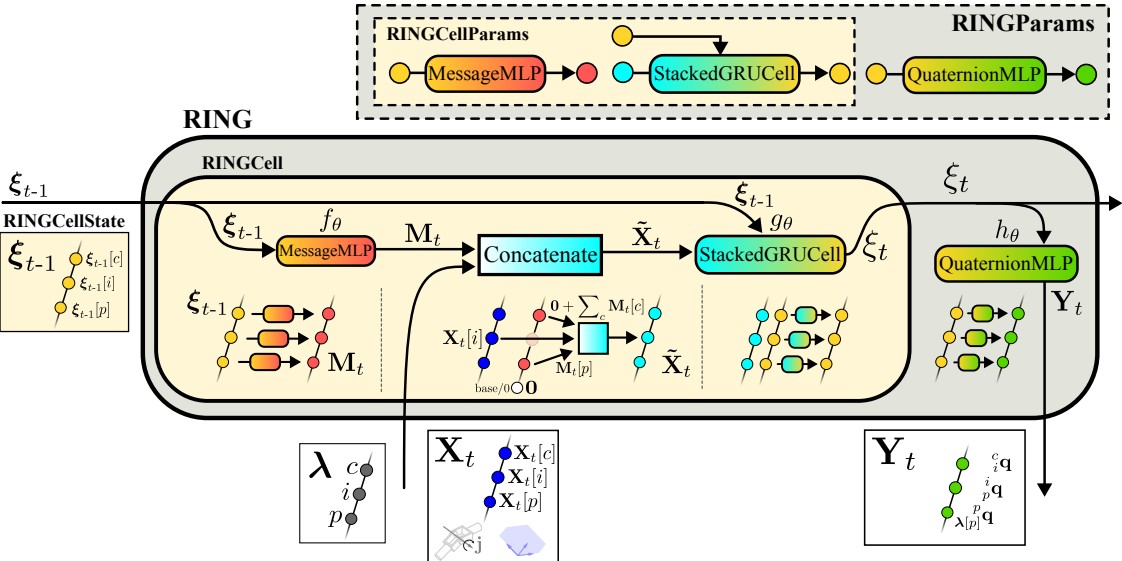

Figure 5: The architecture of the plug-and-play IMT solution RING. It consists of the RNN cell RINGCell and a final MLP head that returns one quaternion per node in the graph. RING is trained to estimate child-to-parent orientations from the available local IMU and joint axis data and nearest neighbour messages. To this end, RINGCell applies a shared set of parameters on a decentralized, per-node level and passes messages along the edges of the graph to allow for information exchange. Note that while the parameters are shared, the hidden states are not shared across nodes. This architecture allows RING to apply a single set of parameters across a broad range of IMTPs, which may vary in aspects such as the number of segments, and it ultimately enables RING's remarkable pluripotency.

### 4.3 The Architecture of RING

RING is based on a decentralized network of message-passing RNNs which allows for information exchange along the edges of a graph (as given by the CG). RING can be applied to an arbitrary CG (see Section 3.3) and it maps per-node-input to per-node-output. *Most importantly, the parameters of RING are shared across the nodes of the graph such that the number of parameters of RING does not depend on the given CG.* The hidden state, however, is not shared across the nodes. This allows for the training of a single RING network which then solves a variety of IMTPs (see Section 4.1).

The introduction of RING stems from the requirement of acquiring a single pretrained network that can solve different IMTPs with a varying number of inputs and outputs, e.g., estimate two orientations for a two-segment KC, and three orientations for a three-segment KC. This is possible because on a node (or segment) level, we can guarantee static input (at most one IMU, at most a known joint axis direction) and output shapes (one orientation relative to parent). But the per-node-output cannot be estimated from only the per-node-input as the orientation relative to the parent depends on the orientation state of the parent, and orientation w.r.t. the base cannot be estimated from 6D IMU data, additionally, the segment may not have an IMU attached. Thus, we have to allow information exchange between nodes and propose a scheme which again gives rise to local static input and output shapes of messages. Finally, we claim that the estimation of the entire pose in a hierarchical approach is inherently natural and provides an advantageous structural prior to subsequent parameter learning which we explore in Section 5.4.

RING consists of a novel RNN cell, named RINGCell, and a final Multi-Layer Perceptron (MLP) head such that the network's per-node-output is a single unit quaternion (per timestep). Note that the network head is independent of the RINGCell and may easily be replaced to suit different needs.

Let $\boldsymbol{\lambda}$ be a CG with $N \in \mathbb{N}$ nodes, let $F \in \mathbb{N}$ be the number of input features per node, let $H \in \mathbb{N}$ be the half-hidden state dimensionality, and let $M \in \mathbb{N}$ be the dimensionality of the latent messages passed inside the cell based on the edges in the CG. Then, let $\boldsymbol{\xi}_{t\text{-}1} \in \mathbb{R}^{N \times 2H}$ be the hidden state of the RINGCell from the previous timestep $t - 1$, and let $\mathbf{X}_t \in \mathbb{R}^{N \times F}$ be the $F$ inputs for all $N$ nodes at time $t$. Then, the next hidden state $\boldsymbol{\xi}_t$ is obtained by

$$\boldsymbol{\xi}_t = \texttt{ringCell}\left(\boldsymbol{\xi}_{t\text{-}1}, \mathbf{X}_t, \boldsymbol{\lambda}\right) \quad \forall t \tag{3}$$

with $\boldsymbol{\xi}_0 = \mathbf{0}$.

Internally, a RINGCell has the parameters of

- a Message-MLP-network $f_\theta : \mathbb{R}^H \to \mathbb{R}^M$ (three layers, single hidden layer, hidden layer size $H$, ReLU activations, no final activation), and

- a Stacked-GRUCell-network $g_\theta : \mathbb{R}^{2H} \times \mathbb{R}^{2M+F} \to \mathbb{R}^{2H}$ which consists of the sequence of Gated-Recurrent-Unit(GRU)Cell, LayerNorm, GRUCell (Cho et al., 2014). Note that $\theta$ is symbolic for the whole set of parameters of the Stacked-GRUCell-network and that it is different to the parameters of $f_\theta$.

Note that the two GRUCells each have a hidden state dimensionality of $H$, thus the hidden state of the Stacked-GRUCell-network is of dimensionality of $2H$.

Then, a RINGCell has three consecutive steps, $\forall i = 1 \ldots N$ :

1. Messages $\mathbf{M}_t \in \mathbb{R}^{N \times M}$ are computed.

$$\mathbf{M}_t[i] = f_\theta\left(\boldsymbol{\xi}_{t\text{-}1}[i, H\text{:}]\right) \tag{4}$$

2. Messages are passed and latent input $\tilde{\mathbf{X}} \in \mathbb{R}^{N \times (2M+F)}$ computed.

$$\tilde{\mathbf{X}}[i] = \texttt{concat}\left(\mathbf{M}_t\big[\boldsymbol{\lambda}[i]\big], \mathbf{0} + \sum_{c \in \mu_\lambda(i)} \mathbf{M}_t[c], \mathbf{X}_t[i]\right) \tag{5}$$

where $\mathbf{M}_t\big[\boldsymbol{\lambda}[i]\big]$ is $\mathbf{0}$ if $\boldsymbol{\lambda}[i] = 0$.

3. Hidden state is updated.

$$\boldsymbol{\xi}_t[i] = g_\theta \left( \boldsymbol{\xi}_{t\text{-}1}[i], \tilde{\mathbf{X}}[i] \right) \tag{6}$$

The architecture of RING is finished by piping the hidden state $\boldsymbol{\xi}_t$ through a final network head, the Quaternion-MLP that combines

- a Layernorm, and a MLP-network $h_\theta : \mathbb{R}^H \to \mathbb{R}^4$ (three layers, single hidden layer, hidden layer size $H$, ReLU activations, no final activation).

The Quaternion-MLP has two consecutive steps, $\forall i = 1 \ldots N$ :

1. Unnormalized output $\tilde{\mathbf{Y}} \in \mathbb{R}^{N \times 4}$ is computed.

$$\tilde{\mathbf{Y}}[i] = h_\theta \left( \texttt{layernorm} \left( \boldsymbol{\xi}_t[i, H\mathord{:}] \right) \right) \tag{7}$$

2. Normalize to allow interpretation as unit quaternions. One unit quaternion per node. Final RING output $\hat{\mathbf{Y}}_t \in \mathbb{H}^N$.

$$\hat{\mathbf{Y}}_t[i] = \frac{\tilde{\mathbf{Y}}[i]}{\sqrt{\sum_{j=1}^{4} \tilde{\mathbf{Y}}[i, j]^2}} \tag{8}$$

Note that the normalizing operation, due to its square-root operation, requires special care to allow for successful backpropagation. Also note that the employed loss function, see Section 3.6, is based on a arctan expression, which does not require any special care, in contrast to seemingly equivalent expressions to extract the angle from a quaternion that are based on arccos.

Finally, by combining the equations (3), (7), (8) and unrolling the RNN in time, we can view the entire RING network as a function that maps the timeseries of available input data $\mathbf{X} \in \mathbb{R}^{T \times N \times F}$ and CG $\boldsymbol{\lambda} \in \mathbb{N}^N$ to the timeseries of predicted output data $\hat{\mathbf{Y}} \in \mathbb{H}^{T \times N}$, i.e.

$$\hat{\mathbf{Y}} = \texttt{ring}_\theta \left( \mathbf{X}, \boldsymbol{\lambda} \right) \tag{9}$$

where the parameters of RING are given by the set $\{f_\theta, g_\theta, h_\theta\}$. The hyperparameters of RING are $H \in \mathbb{N}$ and $M \in \mathbb{N}$. Note that the set of parameters of RING is influenced by the hyperparameters of RING and the number of input features per node $F$, but *not* by the GC $\boldsymbol{\lambda}$ or the number of bodies. For example, a single RING network can be used for predicting the orientations of both two-segment or three-segment KCs even though the number of bodies and, consequently, the dimensionality of input and output arrays is different, e.g., $\mathbf{X} \in \mathbb{R}^{T \times 2 \times F}$ compared to $\mathbf{X} \in \mathbb{R}^{T \times 3 \times F}$.

### 4.4 RING: A Single IMT Solution to a variety of IMTPs

In this section, we use the *simulated* training data from Section 4.2 to train a network based on the RINGCell architecture (see Section 4.3). The trained network is then called RING and, with a single set of parameters, RING shows its pluripotency in a range of *experimental* scenarios, from simple single-joint tracking to complex four-segment KCs, without reliance on magnetometers as will be shown in Section 5.

For each of the four KCs $\boldsymbol{\lambda}_N$ (see Definition 4.1), we use the RCMG (see Algorithm 1) to simulate 512 input-output pairs and stack them to create a single batch of training data. This batch of training data is used to update the parameters of the RING network (see Section 4.3), with $H = 400$ and $M = 200$ (total parameter count: $2\,337\,404$) by minimizing the mean-squared-(orientation-estimation)-error, i.e.,

$$\min_\theta \frac{1}{10T} \sum_{N \in \{1,2,3,4\}} \mathop{\mathbb{E}}_{\mathbf{X}, \mathbf{Y} \sim \text{RCMG}(\boldsymbol{\lambda}_N)} \sum_{t=1}^{T} \sum_{i=1}^{N} \texttt{loss} \left( \mathbf{Y}, \texttt{ring}(\mathbf{X}, \boldsymbol{\lambda}_N) \right) [t, i] \tag{10}$$

where $\texttt{RCMG}$ is given by Algorithm 1, $\texttt{loss}$ is given by eq. (2), $\texttt{ring}$ is given by eq. (9), and where the expectation $\mathbb{E}$ is estimated using 512 draws.

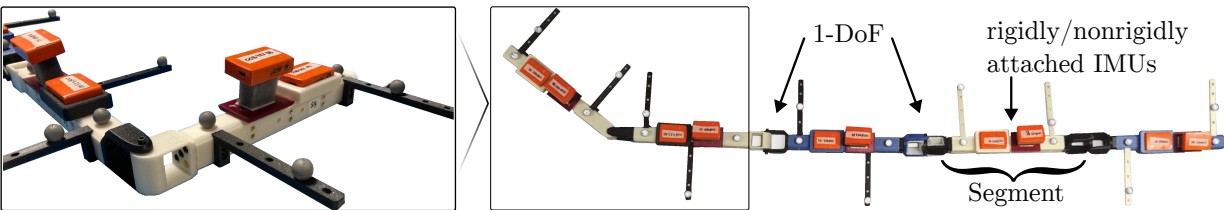

Figure 6: Experimental five-segment KC with ten IMUs (orange boxes) and 20 OMC markers (grey spheres). It uses a single spherical joint followed by three hinge joints, each oriented along the x, y, and z axes, respectively. Each segment of the KC was equipped with two IMUs: one firmly attached to the segment and another affixed nonrigidly using foam padding. The experimental KC is moved in space, and the inertial and optical data is recorded to validate RING and compare it to SOTA methods across a broad range of IMTPs.

We use the LAMB optimizer (You et al., 2020) combined with global (threshold = 0.1) and adaptive (threshold = 0.2) gradient clipping, a cosine decaying learning rate (initial learning rate = $1 \times 10^{-3}$, and Truncated Backpropagation Through Time (TBTT). Borrowing the notation from Williams & Peng (1990), we utilize TBTT(10 s, 10 s), i.e., the gradients are stopped and applied after 10 s instead of the total length of 60 s. This results in every batch generation (or episode) corresponding to six parameter updates. We train for 5000 episodes using a single node with eight A40 GPUs (each 48 gigabytes of VRAM). Due to the large amounts of training data, overfitting is seemingly impossible and any form of early stopping did not prove to be required. This has been also confirmed with a separate validation dataset. Training has been stopped after 5000 episodes as the loss has no longer improved.

For practical applications, the inference time and computational requirements of RING are more important than its training requirements. To this end, a theoretical and empirical analysis of the time complexity of RING is conducted in Appendix D, and the findings are summarized here. The theoretical time complexity to advance the prediction of RING by one timestep is $\mathcal{O}\left(N \times H \times (H + M + F)\right)$. This translates in practice to an efficient NN that enables real-world online application even on low-end hardware. For example, on a single-core Intel Xeon at 2 GHz, RING can comfortably enable motion tracking of a four-segment KC at more than 500 Hz, which is well above typical IMU sampling rates that range from 90 to 286 Hz (Laidig et al., 2021).

### 4.5 Openly-available Code and Data

The code and experimental validation data is hosted at `https://github.com/simon-bachhuber/ring_supplementary_material` and the repository contains implementations of the RCMG, RINGCell, and RING as decoupled components. Additionally, it includes the code of SOTA methods and validation data to create the experimental validation results (AMAEs and RMAEs) of RING and the SOTA methods that are discussed in Section 5. Finally, we also provide files to retrain RING from scratch, without requiring any real-world training data files. The software, most notably, uses the JAX and Haiku frameworks (Bradbury et al., 2018; Hennigan et al., 2020). The ease-of-use of RING is demonstrated through a code example in the Appendix E.

## 5 Experimental Validation of RING

In this section, we evaluate the accuracy of RING with one common set of pretrained parameters (see Section 4.4) across a broad range of *experimental* IMTPs (see Section 4.1). *In general, RING shows remarkable pluripotency in zero-shot generalization to real-world experiments across diverse IMTPs. This underscores its broad applicability.*

In Section 5.1, we describe the experimental setup used to evaluate RING on real-world data. We show that RING successfully solves multiple previously solved challenging IMTPs and competes with the current SOTA methods (Section 5.2). Impressively, RING further achieves accurate tracking in even more challenging IMTPs, including two IMTPs that have not been solved before (Section 5.3).

## 5.1 Experimental Setup and Evaluation Metric

Experimental evaluation is conducted on a five-segment KC which is shown in Figure 6. The KC is constructed by a concatenation of a single spherical joint followed by three hinge joints, each oriented along the x, y, and z axes, respectively. Each segment of the KC was equipped with two IMUs (MTw Awinda, Xsens, Enschede, the Netherlands): one firmly attached to the segment and another affixed nonrigidly using foam padding. This foam-attachment of IMU was conducted to investigate the reduction of motion artifacts. The left panel in Figure 6 shows a close-up of one segment of the KC with one exemplary nonrigidly attached IMU. Each segment is equipped with at least three noncollinearly placed reflective markers that are used to obtain ground truth orientations from a twelve camera Optical Motion Capture (OMC) system (OptiTrack Prime x22, NaturalPoint Inc., Corvalis, USA).

Two distinct trials were conducted, involving random movements of the five-segment KC, by introducing manual alterations of the KC pose by two experienced scientists. The first trial, spanning a duration of $66\,\mathrm{s}$, featured a diverse range of motions, extending from very slow to notably rapid movements. Moreover, the second trial, with a length of $68\,\mathrm{s}$, not only encompassed a variety of motions but also incorporated random intervals of complete stillness, wherein the KC remained motionless.

All trials were preprocessed in the following way: NaN values were removed, an offline time-synchronization was employed via cross-correlation between measured (IMUs) and approximated angular velocities (OMC), and all collected data was resampled to a uniform sampling rate of $100\,\mathrm{Hz}$. Additionally, we correct for any small misalignment between the rigidly attached IMUs' local coordinate systems and the corresponding segments' body coordinate systems as spanned by the OMC markers. Simultaneously, we align the OMC's reference coordinate system with the earth reference coordinate system (the base), as observed by the IMUs. For more comprehensive details regarding these preprocessing steps and the software implementations utilized, readers are referred to the study Laidig et al. (2021).

The rich experimental data obtained from the five-segment KC enables us to perform evaluations on subsets of data and the KC, which represent a variety of different IMTPs. For example, for validation of a one-segment KC, we use the recorded data of all five segments of the five-segment KC independently for evaluation, which effectively increases the amount of validation data by a factor of five. Similarly, for validation of a two-segment KC with a hinge joint, we use three different sub-KCs with a joint axis direction along the x, y, and z axes (the two-segment sub-KC of segment one and segment two is excluded due to its spherical joint). Similarly, for a three-segment KC with double hinge joints, we use two different sub-KCs, and for a four-segment KC with triple hinge joint we use the sub-KC of segment two to five.

Assessing KC pose estimation performance of RING in comparison with existing SOTA and thus solving the IMTP under consideration, requires the usage of a suitable evaluation metric. As already discussed in Section 3.6, the expression $\mathtt{angle}(\mathbf{q} \otimes \hat{\mathbf{q}}^*)$ can be used to compare the difference between a ground truth orientation $\mathbf{q}$ and the corresponding predicted orientation $\hat{\mathbf{q}}$. Thus, in order to compare the timeseries of ground truth $\mathbf{Y} \in \mathbb{H}^{T \times N}$ and predicted pose $\hat{\mathbf{Y}} \in \mathbb{H}^{T \times N}$ of an $N$-segment KC with GC $\boldsymbol{\lambda}_N$, we compute the Attitude-Mean-Absolute-(orientation)-Error (AMAE) and Relative-Mean-Absolute-(orientation)-Error (RMAE) which are given by

$$\mathtt{AMAE}\left(\mathbf{Y}, \hat{\mathbf{Y}}\right) = \frac{1}{T} \sum_{t=500}^{T} \left| \mathtt{angle}\left(\mathtt{zeroHead}\left(\mathbf{Y}[t,1]\right) \otimes \mathtt{zeroHead}\left(\hat{\mathbf{Y}}[t,1]\right)^*\right)\right| \tag{11}$$

$$\mathtt{RMAE}\left(\mathbf{Y}, \hat{\mathbf{Y}}\right) = \frac{1}{TN} \sum_{t=500}^{T} \sum_{i=2}^{N} \left| \mathtt{angle}\left(\mathbf{Y}[t,i] \otimes \hat{\mathbf{Y}}[t,i]^*\right)\right| \tag{12}$$

where $|.|$ denotes the absolute value and $\mathtt{zeroHead}$ removes the heading component (see Definition B.7). Note that initial $5\,\mathrm{s}$ (equaling to an index of 500 at 100 Hertz) of each timeseries were deliberately excluded from the AMAE and RMAE calculations. This decision was made to ensure that the recorded errors accurately reflected the method's performance after convergence.

From Section 3.2, recall that magnetometer-free IMT estimates one absolute attitude, and $N-1$ relative orientations. This difference is captured by the two metrics AMAE and RMAE. Mathematically, AMAE

reports zero angle error if the ground truth orientation and estimated orientation are equal up to an arbitrary heading difference. Whereas, RMAE reports zero angle error if and only if both orientations are exactly identical. The implications are that a low AMAE indicates that the inclination of the system is correctly estimated but the entire system might still be rotated with an arbitrary yaw angle. A low RMAE means that the entire internal pose of the system is accurately estimated.

## 5.2 RING Unifies Prior Work

### 5.2.1 Attitude Estimation

Attitude estimation in real-world environments using inertial sensors is a vital prerequisite for a wide range of applications, including tracking human movement and enabling autonomy in air and land vehicles. The attitude estimation problem is defined as estimating the orientation of an object with respect to the horizontal plane with a single sensor (Weber et al., 2021).

The widely employed and long-standing methods from Madgwick (2010); Mahony et al. (2008); Seel & Ruppin (2017) have recently been outperformed with SOTA approaches by Weber et al. (2021); Laidig & Seel (2023). Table 1 shows that RING aligns with SOTA performance in attitude estimation, as indicated by the experimental AMAEs. More specifically, the AMAE of the attitude for Madgwick (2010) is $(2.25 \pm 0.81)°$, for Laidig & Seel (2023) is $(1.61 \pm 1.04)°$, for Weber et al. (2021) is $(2.06 \pm 1.03)°$, for Mahony et al. (2008) is $(2.09 \pm 0.87)°$, for Seel & Ruppin (2017) is $(2.56 \pm 0.93)°$, and for RING is $(2.13 \pm 0.91)°$.

### 5.2.2 Magnetometer-free Tracking of 1-DoF Joint

IMT of connected segments without relying on magnetometers is desirable for numerous applications where the magnetic field is typically disturbed, including control of industrial robotic manipulators, interconnected drones in automated warehouses, and human motion analysis in hospital environments.

Here, we consider the IMTP of tracking the relative motion between two segments from two 6D IMUs, assuming a single hinge joint with a known axis direction connects both segments.

Overcoming the use of magnetometers is typically achieved by combining SOTA approaches for attitude estimation and utilizing joint-specific constraints that exploit knowledge of the hinge joint axis direction (Laidig et al., 2017; Lehmann et al., 2020). Additionally, a magnetometer-reliant method uses 9D VQF (Laidig & Seel, 2023) for both IMUs independently and does not exploit any kinematic constraint between the two body parts. The RMAEs are reported in Table 1, and they are: for the VQF-based baseline $(19.36 \pm 8.02)°$, for Lehmann et al. (2020) $(4.15 \pm 2.05)°$, for Laidig et al. (2017) $(3.32 \pm 2.12)°$, and for RING $(3.52 \pm 1.00)°$. This shows that RING aligns with the SOTA methods, which are already a non-trivial combination of two separate methods which, in contrast to applying RING, requires expert knowledge.

### 5.2.3 Magnetometer-free Tracking of 1-DoF Joint with Unknown Joint Axis Direction

The IMTP of Section 5.2.2 can be made more challenging by assuming that the hinge joint axis direction is not known. Then, it can first be estimated from the IMU data, and then subsequently a method that assumes a known joint axis direction can be applied.

A SOTA method for hinge joint axis direction estimation is given by Olsson et al. (2020). RING out-of-the-box supports an unknown hinge joint axis direction by its versatility to drop out the respective node input and replacing it with zeros.

The experimental RMAEs are given in Table 1; combining Olsson et al. (2020) with Lehmann et al. (2020) results in $(4.06 \pm 2.23)°$, and with Laidig et al. (2017) results in $(3.18 \pm 2.05)°$, and RING achieves $(3.92 \pm 1.40)°$. This shows that RING aligns with SOTA performance, which *comprises three distinct methods*.

### 5.2.4 Three-Segment Sparse IMT

Only few recent works have achieved the combination of sparse and magnetometer-free IMT. One such challenging IMTP is, the tracking of all relative segment orientations of a three-segment KC, with double

hinge joints and known hinge joint axes directions, by using only two IMUs placed on the outer segments. The SOTA method for this IMTP is given by Bachhuber et al. (2023). The absolute attitude can be easily tracked using any of the methods from Section 5.2.1.

As shown in Table 1, the experimental RMAE using Bachhuber et al. (2023) is $(5.60 \pm 2.35)°$, and using RING the RMAE is reduced to $(4.14 \pm 0.53)°$ and, consequently, RING outperforms the SOTA in this IMTP.

## 5.3 RING Goes Beyond the SOTA

### 5.3.1 Motion Artifact Reduction

The efficacy of all SOTA IMT methods heavily rely on a rigid sensor-to-body attachment. In many practical scenarios, such as human motion analysis, this assumption is rapidly violated, resulting in strongly degraded estimation accuracies, attributed to a model-reality mismatch.

Here, we consider the IMTP of Section 5.2.2, however here, the IMUs are not rigidly attached to the two segments, while the estimation target remains the relative orientation between the two segments (and not between the coordinate systems of the two IMUs), and the absolute attitude of one of the segments.

Currently, there exist no method that does not make the assumption of rigid IMU attachment. Still, the methods from Section 5.2.1 can be applied to estimate the absolute attitude, and we use the most accurate estimator VQF (Laidig & Seel, 2023) for comparison purposes. The methods from Section 5.2.2 are applied to estimate the relative orientation. To compensate for the violation of the rigid-IMU-attachment assumption, we additionally apply an intuitive low-pass filter (LPF) step to the estimated absolute attitude and relative orientation for each baseline method to suppress unwanted high frequency artifacts. The cutoff frequency was grid searched and we report only the best result for each baseline. Alternatively, the use of the LPF on the estimated orientations prior to computing the relative orientation did not yield better performance.

In Table 1, column 5.3.1A reports the experimental AMAE in the attitude estimate, and demonstrates RING's superior performance of $(7.59 \pm 2.85)°$ over the combination of VQF+LPF with an AMAE of $(9.19 \pm 2.31)°$. Similarly, column 5.3.1B reports experimental RMAEs, they are: Lehmann et al. (2020) achieves $(8.00 \pm 2.78)°$, Laidig et al. (2017) achieves $(7.00 \pm 1.57)°$, and RING achieves $(5.56 \pm 2.33)°$. This shows that RING outperforms the SOTA methods. Note that the reduced AMAE shows that RING has learned to fuse the information of the second segment's IMU into the attitude estimation of the first segment.

### 5.3.2 Three-Segment Sparse IMT with Unknown Joint Axes Directions

The IMTP considered in Section 5.2.4 can be made even more challenging by not assuming known joint axes directions.

To the best of the authors' knowledge, there currently exists no method that is applicable in such a challenging IMTP. RING can solve this IMTP with only a modest increase in error (Table 1), given the increased complexity of the task. Note that the direction of the joint axis cannot be estimated using Olsson et al. (2020) such that the method from Section 5.2.4 may be applied, as Olsson et al. (2020) does not allow for a sparse sensor setup which inherently requires a pluripotent approach such as RING. We report a RMAE value of $(5.37 \pm 0.71)°$ for RING in this challenging IMTP.

### 5.3.3 Four-Segment Sparse IMT: 3-DoFs between IMUs

Increasing sensor sparsity will naturally make an IMTP problem more complex. A KC configuration with four segments and only two 6D IMUs results in three DoFs (three consecutive hinge joints) between the two outer-segment 6D IMUs and it represents the limit of accurate sparse IMT. Despite the complexity, the estimation target remains to capture all three relative orientations.

To the best of the authors' knowledge, there currently is no method that is applicable in such a challenging IMTP. RING can solve this problem formulation sufficiently well. When assuming known joint axes directions for the three hinge joints, RING achieves a RMAE of $(6.78 \pm 1.41)°$. An exemplary trial is shown in Figure 7.

Table 1: Motion tracking accuracy (in degrees) of RING compared to various SOTA methods across a variety of IMTPs. While previous methods are problem-specific and Not Applicable (NA) to many IMTPs, RING is the only method that accurately solves all problems. All columns report the RMAE, see eq. (12), as metric, except for the columns 5.2.1 and 5.3.1A which report AMAE as metric, see eq. (11).

| IMTPs Method | 5.2.1 | 5.2.2 | 5.2.3 | 5.2.4 | 5.3.1a | 5.3.1b | 5.3.2 | 5.3.3 |
|---|---|---|---|---|---|---|---|---|
| | 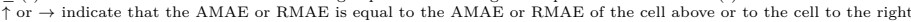 | | | | | | | |
| (1) | $2.06 \pm 1.03$ | NA | NA | NA | $\geq(5)$ | NA | NA | NA |
| (2) | $2.25 \pm 0.81$ | $\geq(5)$ | $\geq(5)$ | NA | $\geq(5)$ | NA | NA | NA |
| (3) | $2.09 \pm 0.87$ | $\geq(5)$ | $\geq(5)$ | NA | $\geq(5)$ | NA | NA | NA |
| (4) | $2.56 \pm 0.93$ | $\geq(5)$ | $\geq(5)$ | NA | $\geq(5)$ | NA | NA | NA |
| (5) | $1.61 \pm 1.04$ | $\rightarrow$ | $19.3 \pm 8.02$ | NA | $9.20 \pm 2.31$ | $24.9 \pm 17.6$ | NA | NA |
| (5)+(6) | $\uparrow$ | $3.32 \pm 2.12$ | NA | NA | $\uparrow$ | $7.00 \pm 1.57$ | NA | NA |
| (5)+(7) | $\uparrow$ | $4.15 \pm 2.05$ | NA | NA | $\uparrow$ | $8.00 \pm 2.78$ | NA | NA |
| (5)+(6)+(8) | $\uparrow$ | $\rightarrow$ | $3.18 \pm 2.05$ | NA | $\uparrow$ | $8.50 \pm 2.60$ | NA | NA |
| (5)+(7)+(8) | $\uparrow$ | $\rightarrow$ | $4.06 \pm 2.23$ | NA | $\uparrow$ | $7.90 \pm 2.48$ | NA | NA |
| (9) | NA | NA | NA | $5.60 \pm 2.35$ | NA | NA | NA | NA |
| RING | $2.13 \pm 0.91$ | $3.52 \pm 1.00$ | $3.92 \pm 1.40$ | $4.14 \pm 0.53$ | $7.59 \pm 2.85$ | $5.56 \pm 2.33$ | $5.37 \pm 0.71$ | $6.78 \pm 1.41$ |

Methods: Weber et al. (2021)(1), Madgwick (2010)(2), Mahony et al. (2008)(3), Seel & Ruppin (2017)(4), Laidig & Seel (2023)(5), Laidig et al. (2017)(6), Lehmann et al. (2020)(7),Olsson et al. (2020)(8), Bachhuber et al. (2023)(9)
$\geq (i)$ refers to the AMAE or RMAE of being expected to be larger or equal than for method $(i)$
$\uparrow$ or $\rightarrow$ indicate that the AMAE or RMAE is equal to the AMAE or RMAE of the cell above or to the cell to the right

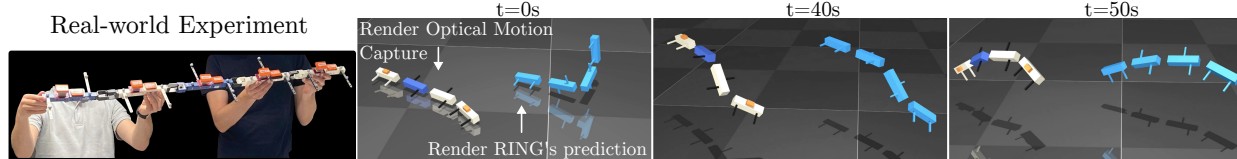

Figure 7: Exemplary frames that showcase RING's performance for the IMTP that involves a four-segment KC with sparse IMU attachments and known joint axes directions (see Section 5.3.3). It is a remarkable first that RING can accurately estimate the four orientations (one absolute and three relative orientations) from only two magnetometer-free IMUs. A video of the trial is available here.

When assuming unknown joint axes directions, RING achieves $(13.66 \pm 3.07)°$. This shows that with unknown joint axes directions, this IMTP pushes the limits of observability (Bachhuber et al., 2022).

## 5.4 The Decentralized Approach of RING Provides an Advantageous Structural Prior

In this section, we showcase a second advantage of the decentralized approach of RING over a centralized approach that is typically employed, e.g., by stacking multiple LSTM- or GRU-Cells. First, recall that RING is based on a decentralized network of message-passing RNNs which allows for the training of a single set of parameters despite a varying number of bodies in the KC and, while the latter aspect is the core motivation behind this architecture it is, interestingly, not the sole motivation behind the decentralized approach. The second motivation is that the estimation of the entire pose in RING's decentralized approach provides an advantageous structural prior compared to an RNN that utilizes a centralized approach. For this purpose, we compare RING to the RNN-based Observer (RNNO), a deep GRU network with intermediate Layernorm layers, as it was proposed in Bachhuber et al. (2023). RNNO maps all available input data to the entire targeted pose data and does not utilize the specific graph structure of the IMTP. RNNO has been proposed as the solution of a specific IMTP (see Section 5.2.4) and, for this IMTP, both RING and RNNO achieve similar performance. However, this is not the case if the IMTP becomes vastly more complex. Consider, e.g., the IMTP of Section 5.3.3, which is arguably the most challenging IMTP under consideration in this work. We have trained RNNO on the subset of training data of RING that corresponds to this IMTP. Despite

Table 2: Motion tracking accuracy of RING solving the IMTP defined in Section 5.2.3 on external datasets. RING provides consistently low errors demonstrating robustness across different IMU hardware.

| Dataset | IMU Hardware | T[s] | Native Rate [Hz] | RMAE [°] |
|---|---|---|---|---|
| Ours | Xsens MTw Awinda | 402 | 40 | $3.92 \pm 1.40$ |
| RepoIMU (1) | microIMU (2) | 390 | 90 | $3.44 \pm 0.06$ |
| OpenAXES (3) | Bosch BMI160 + Analog Devices ADXL355 | 227 | $\approx 125$ | $2.52 \pm 0.29$ |

Szczęsna et al. (2016) (1), Jędrasiak et al. (2013) (2), Webering et al. (2023) (3)

varying several hyperparameters, the best reported RMAE of RNNO is $(18.72 \pm 5.12)°$. This is significantly higher compared to RING's RMAE of $(6.78 \pm 1.41)°$, even though RING is maintaining its applicability to a broad range of IMTPs, and is not purpose-trained for a single IMTP. This showcases that the decentralized approach of RING provides an advantage even if only the solution of a single, specific IMTP is required.

## 5.5 RING's Robustness to Different IMUs

IMUs in general, and especially across vendors, differ in properties like noise density and bias offset. RNN-based inertial sensor fusion has been demonstrated to generalize across different sensor hardware (Weber et al., 2021). Nonetheless, to ensure broad real-world applicability, we investigate RING's robustness to different IMU hardware. First, we analyze how RING's performance scales as noise and bias properties are incrementally increased. Then, we evaluate RING on openly-available, real-world datasets that use IMUs from different vendors.

To simulate reasonable noise density and bias offset ranges, we constructed a worst-case IMU by combining the worst properties of various IMU manufacturers, as summarized in Table 3. Then, those worst-case noise and bias values are incrementally (in seven equidistant steps) increased from 0 to 120%, and a corresponding amount of simulated noise and bias is added to our real-world IMU data (see Section 5.1), which yields seven modified datasets. RING and all SOTA methods are validated on the modified dataset, and this procedure is repeated using ten different seeds. The AMAEs and RMAEs of all methods are plotted for all IMTPs as a function of the seven steps in Figure 9. The figure shows that, whilst the performance of all methods (as expected) slightly worsens as noise and bias are increased, RING maintains accuracy comparable to SOTA methods in all IMTPs and, especially in two-segment KC tracking, substantially outperforms them.

The RepoIMU dataset (Szczęsna et al., 2016) contains real-world IMU and ground truth data from passive motions of a swinging pendulum. It uses non-commercial micro IMUs (Jędrasiak et al., 2013) with a focus on low cost and small size over accuracy. The OpenAXES Robot Dataset contains data from motions of a robot arm drawing different shapes at different speeds. IMUs are attached to each segment, and the ground truth is known from the robot's encoders. Table 2 reports RING's performance for both external datasets.

## 6 Discussion

We have shown that a single RNN, named RING, can accurately solve a broad range of IMTPs even if two IMTPs do not have same input-output dimensionality. This pluripotent behavior originates from a decentralized architecture that provides an advantageous structural prior compared to the more typical centralized approach which, in contrast to RING, has the additional limitation that it can only be utilized if the solution of a single IMTP is sufficient. In summary, for the set of IMTPs under consideration (see Section 4.1), we have shown that RING can enable accurate IMT for one-, two-, and three-segment KCs as long as the IMUs are rigidly attached. Most notably, this includes an IMTP that is sparse, magnetometer-free, and requires sensor-to-segment calibration. If IMUs are nonrigidly attached, then RING is shown to provide accurate orientation estimates for a two-segment KC. RING can achieve accurate IMT for a four-segment KC, provided that the IMUs are rigidly attached and sensor-to-segment calibration is not required. It is a remarkable first, that RING can track the pose of the four-segment KC (which requires in total four distinct orientations) using only two IMUs. Note that RING does not require problem specific priors, although their

introduction, i.e., a known joint axes direction, or an additionally attached IMU is straightforwardly possible as additional input data to improve the motion tracking accuracy even further. This can be seen in Table 1 where, e.g., for a three-segment KC, providing joint axes directions decreases the mean RMAE from 5.37° (Section 5.3.2) to 4.14 (Section 5.2.4).

## 7 Conclusion

In the presented work, we combined ideas from the domain of multi-agent systems with RNNs to propose an architecture based on a decentralized network of message-passing, parameter-sharing RNNs. We have successfully exploited this combination for the analysis of structural sequential data and solved a challenging state estimation problem, which is IMT of KCs, by letting the decentralized network map local IMU measurements and nearest-neighbour messages to local orientations. In particular, we introduced RING, a *pluripotent* IMT solution that, unlike all previous, problem-specific approaches, enables plug-and-play non-expert use. RING outperforms a range of problem-specific SOTA solutions and even generalizes to previously unsolved scenarios, including the challenging combination of magnetometer-free and sparse sensing with unknown sensor-to-segment parameters. Remarkably, RING demonstrates the ability to zero-shot generalize to experimental scenarios, despite being trained solely on simulated data. For example, RING can, for the first time, accurately track a real-world four-segment kinematic chain (which requires estimating four orientations) using only two magnetometer-free IMUs.

RING's pluripotency greatly simplifies the application of IMT by eliminating the need for expert knowledge to identify, select, and fine-tune problem-specific methods. This is expected to not only make IMT more powerful and less restrictive in established domains but also to facilitate the accessibility of IMT technology by non-expert users and broadens its applicability to previously untapped domains.

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

# A Preliminaries

## A.1 Notation

- Scalars are lowercase or uppercase, italic, non-bold, e.g., $x \in \mathbb{R}$, or, e.g., $N \in \mathbb{N}$.

- (Column) vectors are lowercase, italic, bold, e.g., $\boldsymbol{x} \in \mathbb{R}^3$, and, e.g., $\boldsymbol{x} = (1,2,3)^\intercal$.

- Matrices (or higher) are uppercase, upright, bold, e.g., $\mathbf{X}$ and $\mathbf{X} \in \mathbb{R}^{3\times4}$.

- Individual quaternions are denoted with $\mathbf{q} \in \mathbb{H}$. One-dimensional arrays of quaternions with $\mathbf{q}$, e.g., $\mathbf{q} \in \mathbb{H}^5$. Two dimensional arrays of quaternions with $\mathbf{q}$, e.g., $\mathbf{q} \in \mathbb{H}^{5\times5}$.

- (Programming) functions and structures are written in typewriter typestyle, e.g., `sys`.

- The equal symbol $=$ is overloaded and used for definitions, assignments, and comparison, and the context defines the current meaning.

- The symbol $\mathbf{0}$ defines an arbitrarily large array of zeros that is automatically broadcasted to the required dimensionality.

- The symbol $\mathbb{1}$ defines either the unity element of a given space, or the indicator function, such that, e.g., $\mathbb{1}_0(i)$ is one for $i = 0$ and zero else.

- The symbol $\otimes$ is used to denote the direct product of vector spaces, and additionally denotes quaternion multiplication, see Definition B.2.

## A.2 Array Indexing and Slicing

Vectors and matrices are array-like objects that can be indexed and sliced dynamically. Indexing starts with 1 and slicing is inclusive on both sides. For example, let $\mathbf{X} \in \mathbb{R}^{3\times4}$, then $\mathbf{X}[1] \in \mathbb{R}^4$, or $\mathbf{X}[1{:}2] \in \mathbb{R}^{2\times4}$.

Additionally, we define the following auto-completion rules by example, such that $\mathbf{X}[{:}2]$ is equivalent to $\mathbf{X}[1{:}2]$, and $\mathbf{X}[2{:}]$ is equivalent to $\mathbf{X}[2{:}3]$, and $\mathbf{X}[{:}]$ is equivalent to $\mathbf{X}[1{:}3]$. Finally, multiple dimensions can be indexed or sliced simultaneously and are separated by a comma, e.g., $\mathbf{X}[{:},3{:}] \in \mathbb{R}^{3\times2}$.

# B Quaternion Algebra

**Definition B.1.** We use $\mathbb{H}$ to denote the space of all unit quaternions, and we denote a unit quaternion with $\mathbf{q} = q_w + q_x i + q_y j + q_z k$.

**Definition B.2.** Let $\mathbf{q}_1, \mathbf{q}_2 \in \mathbb{H}$ be two unit quaternions, then we use $\otimes$ to denote quaternion multiplication of the two unit quaternions, that is

$$
\begin{aligned}
\mathbf{q}_1 \otimes \mathbf{q}_2 = {} & (q_{1w}q_{2w} - q_{1x}q_{2x} - q_{1y}q_{2y} - q_{1z}q_{2z}) + (q_{1w}q_{2x} + q_{1x}q_{2w} + q_{1y}q_{2z} - q_{1z}q_{2y})i \\
& + (q_{1w}q_{2y} - q_{1x}q_{2z} + q_{1y}q_{2w} + q_{1z}q_{2x})j + (q_{1w}q_{2z} + q_{1x}q_{2y} - q_{1y}q_{2x} + q_{1z}q_{2w})k
\end{aligned}
$$

Note that the space of unit quaternions $\mathbb{H}$ in combination with quaternion multiplication $\otimes$ forms a closed group, i.e., $(\mathbf{q}_1 \otimes \mathbf{q}_2) \in \mathbb{H} \quad \forall \mathbf{q}_1, \mathbf{q}_2$.

**Definition B.3.** The inverse of a quaternion $\mathbf{q}^{-1}$ is given by the complex conjugate denoted by $\mathbf{q}^*$.

**Definition B.4.** The quaternion that corresponds to a certain rotation around an axis $\boldsymbol{j} = (j_x, j_y, j_z)^\intercal \in \mathbb{R}^3$ by an angle $\alpha \in \mathbb{R}$ is given by $\mathbf{q} = \texttt{quat}(\boldsymbol{j}, \alpha) = \cos(\frac{\alpha}{2}) + (j_x \sin(\frac{\alpha}{2}))i + (j_y \sin(\frac{\alpha}{2}))j + (j_z \sin(\frac{\alpha}{2}))k$.

**Definition B.5.** Extracting the angle $\alpha$ from a given quaternion $\mathbf{q}$ (the inverse operation of B.4) can be done using $\texttt{angle}(\mathbf{q}) = 2 \arctan\left(\frac{\sqrt{q_x^2 + q_y^2 + q_z^2}}{q_w}\right)$.

**Definition B.6.** We define the projection `project` of a quaternion $\mathbf{q}$ onto a (primary) axis $\mathbf{J}$ as the decomposition into two quaternions, the primary rotation $\mathbf{q}_p$ and the residual rotation $\mathbf{q}_r$, such that $\mathbf{q} = \mathbf{q}_r \otimes \mathbf{q}_p$ while the angle of the residual rotation is minimized. This can be done with

1. $\alpha_p \leftarrow \arctan\left(\frac{j_x q_x + j_y q_y + j_z q_z}{q_w}\right)$

2. $\mathbf{q}_p \leftarrow \texttt{quat}(\boldsymbol{j}, \alpha_p)$

3. $\mathbf{q}_r \leftarrow \mathbf{q} \otimes \mathbf{q}_p^*$

**Definition B.7.** We define the function `zeroHead` as the function that maps a quaternion $\mathbf{q}$ to the quaternion $\mathbf{q}_r$ with zero heading component and returns $\mathbf{q}_r$. It can be obtained via

1. $\mathbf{q}_p, \mathbf{q}_r \leftarrow \texttt{project}\left(\mathbf{q}, (0,0,1)^\intercal\right)$

where `project` is given in Definition B.6.

**Definition B.8.** The function `randQuat` that returns a random quaternion, uniform on the sphere, can be obtained by drawing i.i.d. four numbers from a normal distribution, interpreting them as the components $q_w, q_x, q_y, q_z$ of a quaternion, and then normalizing the quaternion to obtain a unit quaternion.

**Definition B.9.** The function `rotate`$(\mathbf{q}, \boldsymbol{r})$ applies a quaternion $\mathbf{q}$ to a vector $\boldsymbol{r} \in \mathbb{R}^3$. If the quaternion is interpreted as ${}^0_1\mathbf{q}$ (from 0 to 1) and the vector is expressed using the unit-vectors of coordinate system 0, then the function `rotate` returns the same vector but using the unit-vectors of coordinate system 1. Let $\boldsymbol{r} = (r_x, r_y, r_z)^\intercal$, then the `rotate`$(\mathbf{q}, \boldsymbol{r})$ function is given by $(\mathbf{q} \otimes (0, r_x, r_y, r_z)^\intercal \otimes \mathbf{q}^*)\,[1{:}]$.

## C  Training Data: The RCMG Algorithm

---
**Algorithm 2** `randSys` (RCMG First Step)

---
1: **Input:** $\boldsymbol{\lambda}_N$
2: **Output:** sys
3: sys $\leftarrow$ `initSys`$(\boldsymbol{\lambda}_N)$ {allocate empty structure}
4: sys $\leftarrow$ `randBase`$(\text{sys}, \boldsymbol{\lambda}_N)$ {see Definition C.1}
5: **for** $i = 1$ **to** $N$ **do**
6:     sys.$\mathbf{J}[i] = $ `rotate`$(\texttt{randQuat}(), \hat{\boldsymbol{e}}_x)$ {random hinge joint axis direction; unused if sys.$\boldsymbol{\lambda}[i] = 0$}
7:     **for** $d = 1$ **to** $3$ **do**
8:         sys.$\mathbf{R}[i, d] = $ `randSegmentToSegment`$(d)$ {see Definition C.2}
9:         sys.$\mathbf{R}[i+N, d] = $ `randSensorToSegment`$(d)$ {see Definition C.2}
10:     **end for**
11:     {IMU of node 1 of the standard system is always rigidly attached, as there is no second IMU whose measurements may be fused to effectively eliminate motion due to the nonrigid attachment.}
12:     **if** sys.$\boldsymbol{n}[1] = i$ **or** `randBernoulli`$(0.25)$ **then**
13:         $\boldsymbol{k} \leftarrow$ `getRigidStif`$()$ {see Definition C.4}
14:         $\boldsymbol{\gamma} \leftarrow$ `getRigidDamp`$()$ {see Definition C.4}
15:     **else**
16:         $\boldsymbol{k} \leftarrow$ `randNonRigidStif`$()$ {see Definition C.5}
17:         $\boldsymbol{\gamma} \leftarrow$ `randNonRigidDamp`$()$ {see Definition C.5}
18:     **end if**
19:     sys.$\mathbf{K}[i] = \boldsymbol{k}$
20:     sys.$\boldsymbol{\Gamma}[i] = \boldsymbol{k} \cdot \boldsymbol{\gamma}$ {element-wise multiplication}
21: **end for**

---

---

**Algorithm 3** `randMotion` (RCMG Second Step)

---

1: **Input:** `sys`, (`motionConfig`) {`motionConfig` object is used only by `randFreeTraj` and `randHingTraj` and it influences and constraints the random motion (see Appendix C.1)}
2: **Output:** $\mathbf{T} \in \left(\mathbb{H} \otimes \mathbb{R}^3\right)^{2N \times T}$
3: $T \leftarrow \text{int}\left(\frac{60}{\text{sys.T}_s}\right)$ {# timesteps; duration of training sequences is 60 seconds}
4: $\mathbf{Q} = \mathbf{0}$ {timeseries of minimal coordinates $\mathbf{Q} \in \mathbb{R}^{N_q \times T}$ of system without IMU bodies; see Definition 3.2}
5: $a \leftarrow 0$
6: **for** $i = 1$ **to** $N$ **do**
7:    **if** `sys.`$\boldsymbol{\lambda}[i] = 0$ **then**
8:       $b \leftarrow a + 7$
9:       $\mathbf{Q}[a{:}b] = \text{randFreeTraj}\left(\text{motionConfig}, T\right)$ {see Definition C.6}
10:    **else**
11:       $b \leftarrow a + 1$
12:       $\mathbf{Q}[a{:}b] = \text{randHingTraj}\left(\text{motionConfig}, T\right)$ {see Definition C.6}
13:    **end if**
14:    $a \leftarrow b$
15: **end for**
16: $\tilde{\mathbf{Q}} = \mathbf{0}$ {timeseries of minimal coordinates $\tilde{\mathbf{Q}} \in \mathbb{R}^{T \times (N_q + 7N)}$ of system with IMU bodies; see Definition 3.2}
17: $\tilde{\boldsymbol{q}} \leftarrow \mathbf{0}$
18: $\dot{\tilde{\boldsymbol{q}}} \leftarrow \mathbf{0}$ {$\dot{\tilde{\boldsymbol{q}}} \in \mathbb{R}^{N_q + 6N}$; see Definition 3.2}
19: **for** $t = 1$ **to** $T$ **do**
20:    $\boldsymbol{\tau} \leftarrow \text{PDControl}\left(\text{sys}, \mathbf{Q}[:, t], \tilde{\boldsymbol{q}}[:N_q]\right)$ {see Definition C.7}
21:    $\boldsymbol{\tau} \leftarrow \text{concat}\left(\boldsymbol{\tau}, \mathbf{0} \in \mathbb{R}^{6N}\right)^{\mathsf{T}}$ {$N$ IMUs' *passive* free joints}
22:    $\tilde{\boldsymbol{q}}, \dot{\tilde{\boldsymbol{q}}} \leftarrow \text{forDyn}\left(\text{sys}, \tilde{\boldsymbol{q}}, \dot{\tilde{\boldsymbol{q}}}, \boldsymbol{\tau}\right)$ {see Definition C.8}
23:    $\tilde{\mathbf{Q}}[t] = \tilde{\boldsymbol{q}}$
24: **end for**
25: $\mathbf{T} \leftarrow \text{forKin}\left(\text{sys}, \tilde{\mathbf{Q}}\right)$ {see Definition C.8}

---

---

**Algorithm 4** `getXY` (RCMG Third Step)

---

1: **Input:** `sys`, $\boldsymbol{\lambda}_N$, $\mathbf{T} \in \left(\mathbb{H} \otimes \mathbb{R}^3\right)^{2N \times T}$
2: **Output:** $\mathbf{X} \in \mathbb{R}^{N \times 9 \times T}$, $\mathbf{Y} \in \mathbb{H}^{N \times T}$
3: $\mathbf{X} \leftarrow \mathbf{0}$
4: $\mathbf{Y} \leftarrow \mathbf{0}$
5: **for** $i = 1$ **to** $N$ **do**
6:     $p \leftarrow \boldsymbol{\lambda}_N[i]$
7:     $\tilde{i} \leftarrow$ `sys.`$\boldsymbol{n}[i]$
8:     $\tilde{p} \leftarrow 0$
9:     **if** $p \neq 0$ **then**
10:       $\tilde{p} \leftarrow$ `sys.`$\boldsymbol{n}[p]$
11:     **end if**
12:     $O_i \leftarrow$ `isOuter`$(i, $ `sys.`$\boldsymbol{\lambda})$ {true if body $i$ is an outer body; see Definition 3.1}
13:     **if** $O_i$ **or** `randBernoulli`$(0.33)$ {inner IMU data might not be made available} **then**
14:       $j \leftarrow \tilde{i} + N$ {body number IMU node}
15:       $\mathbf{X}[i, {:}6] =$ `simIMU`$\left(\mathbf{T}[j], $ `sys.T`$_{\mathsf{s}}\right)$ {see Definition C.9}
16:     **end if**
17:     **if** $p \neq 0$ **and** `randBernoulli`$(0.5)$ {for hinge joints, joint axis might not be made available} **then**
18:       $\mathbf{X}[i, 7{:}] =$ `sys.J`$[\tilde{i}]$
19:     **end if**
20:     ${}_p^0\mathbf{q} \leftarrow \mathbb{1}$
21:     **if** $\tilde{p} \neq 0$ **then**
22:       ${}_p^0\mathbf{q} \leftarrow \mathbf{T}[\tilde{p}, {:}4]$
23:     **end if**
24:     ${}_i^0\mathbf{q} \leftarrow \mathbf{T}[\tilde{i}, {:}4]$
25:     ${}_p^i\mathbf{q} \leftarrow {}_p^0\mathbf{q} \otimes {}_i^0\mathbf{q}^*$ {note that the expression ${}_p^0\mathbf{q}^* \otimes {}_i^0\mathbf{q}$ can not be used instead of ${}_p^0\mathbf{q} \otimes {}_i^0\mathbf{q}^*$, as it can dramatically reduce the network's ability to learn. The reason is that in the expression ${}_p^0\mathbf{q} \otimes {}_i^0\mathbf{q}^*$ the joint axis direction is expressed in the (more meaningful) local coordinate system and not in the base's coordinate system.}
26:     **if** $\tilde{p} = 0$ **then**
27:       ${}_p^i\mathbf{q} \leftarrow$ `zeroHead`$\left({}_p^i\mathbf{q}\right)$ {see Definition B.7}
28:     **end if**
29:     $\mathbf{Y}[i] = {}_p^i\mathbf{q}$
30: **end for**

---

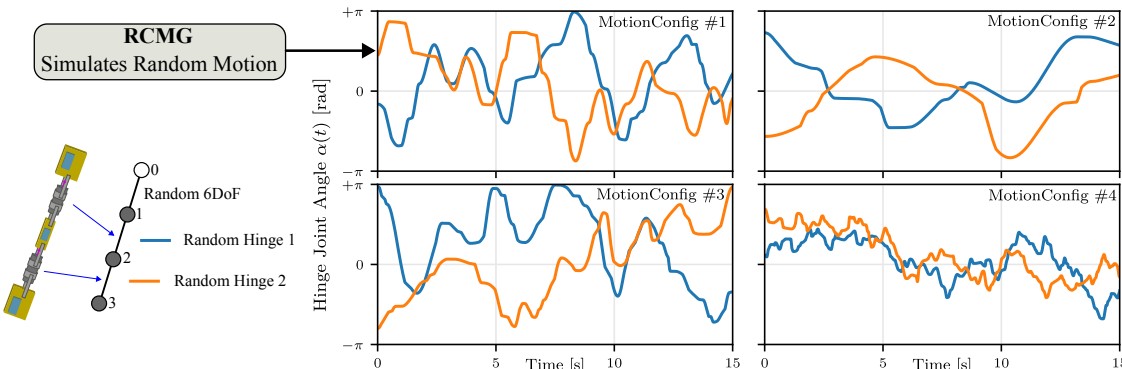

Figure 8: RCMG generates random motion by drawing random trajectories of the minimal coordinates of the system. Exemplary random trajectories of the hinge joint's minimal coordinates are shown for the four different `motionConfig`s (see Appendix C.1). They are drawn using the function `randHingeTraj`, see Definition C.6. RCMG also draws the random trajectory of the minimal coordinates of the free joint, but they are not shown here for simplicity.

## C.1  The `motionConfig` Object

In the second step of the RCMG (see Section 4.2.2), random motion of the KC is simulated and, subsequently, training data is generated. Random motion of the KC is obtained by using PD control during a dynamical forward simulation such that the minimal coordinates of the KC track a randomly drawn set of reference trajectories for the minimal coordinates. The functions `randFreeTraj` and `randHingTraj` (see Definition C.6) are used to draw the minimal coordinates reference trajectories in Algorithm 3. Furthermore, the type of motion generated by these functions can be manipulated with a `motionConfig` object which defines various parameters, e.g., upper limits on angular velocities or lower limits on the amount of motion. For an exhaustive list of parameters, the reader is referred to the software implementation (see Appendix 4.5).

In this work, we use in total four different `motionConfig`s to generate training data. For each generated sequence, we randomly and uniformly draw from these four. The four `motionConfig`s are used to ensure that a wide range of different motion patterns are covered in the training data. Exemplary trajectories of the hinge joints' minimal coordinates are shown in Figure 8.

## C.2  Support Functions used in Algorithms 2/3/4

In the following, the functions used in Algorithms 2/3/4 will be discussed. And while no pseudo-code is provided for these support functions, it should be noted that the majority of these functions should be understandable despite a textual description only. Additionally, the reader may always refer to the software implementation for additional details (see Appendix 4.5).

**Definition C.1.** The function `randBase(sys, `$\boldsymbol{\lambda}_N$`)` randomly re-attaches the base of the system `sys` and afterwards the nodes in the graph are re-numbered according to Section 3.3. This process yields a new parent array $\boldsymbol{\lambda}$. Additionally, the permutation of the new numbers of the nodes expressed in the numbering scheme that was used to obtain $\boldsymbol{\lambda}_N$ is captured in the numbering array $\boldsymbol{n}$.

Consider the three-segment KC given in Figure 2. The parent array is given by $\boldsymbol{\lambda}_3 = (0, 1, 2)^\intercal$ and we assume, without loss of generality, the numbering scheme that is shown in the figure. Here, the function `randBase` has three choices for attaching the base. The first is trivial and given by node 1. The second is given by node 2. In this case the parent array is always given by $\boldsymbol{\lambda} = (0, 1, 1)^\intercal$, and two scenarios for the numbering array are possible. They are $\boldsymbol{n} = (2, 1, 3)^\intercal$ and $\boldsymbol{n} = (3, 1, 2)^\intercal$. The third and last option is given by node 3. In this case the parent array is unchanged but $\boldsymbol{n} = (3, 2, 1)^\intercal$.

A second example is given in Figure 3.

**Definition C.2.** The functions `randSegmentToSegment` and `randSensorToSegment` randomize the body-to-body- and sensor-to-body positions expressed in the local coordinate, respectively. Internally, they both draw these vectors randomly from uniform values ranges. These value ranges are chosen such that there exists a dominant longitudinal direction (x-component), and two equal transversal directions.

**Definition C.3.** The function `randBernoulli`($p$) returns 1 (or true) with probability $p$ and zero (or false) else.

**Definition C.4.** The functions `getRigidStif` and `getRigidDamp` return the stiffness and damping parameters (both are six-dimensional) of the free joint that connects body $i$ to IMU body $i + N$. This mechanism is used to simulate nonrigid IMU attachment. However, these functions provide a fixed set of values that leads to highly stiff and critically damped spring-damper system such that there is effectively no motion between body $i$ and IMU body $i + N$.

**Definition C.5.** The functions `randRigidStif` and `randRigidDamp` return the stiffness and damping parameters of the free joint that connects body $i$ to IMU body $i + N$. Internally, both functions draw these six-dimensional vector randomly from log-uniform values ranges.

**Definition C.6.** The function `randFreeTraj`(motionConfig, $T$) returns a random trajectory of minimal (for free joint minimal=maximal) coordinates $\in \left(\mathbb{H} \otimes \mathbb{R}^3\right)^T$ with $T$ timesteps for a free 6-DoF joint. The function `randHingeTraj`(motionConfig, $T$) returns a random trajectory of minimal coordinates $\in \mathbb{R}^T$ with $T$ timesteps for a hinge joint. Internally, they both use the `motionConfig` (see Appendix C.1) to constraint the random motion to physically relevant motion, i.e., the, e.g., angular velocity is bounded from above. Exemplary trajectories are shown in Figure 8. For additional details, the reader is referred to the software implementation, see Appendix 4.5.

**Definition C.7.** The function `PDControl`(sys, $\boldsymbol{q}_r, \boldsymbol{q}$) computes the generalized force vector $\boldsymbol{\tau} \in \mathbb{R}^{N_{\dot{q}}}$ using a decentralized scheme using $N$ independent PD controllers, and where $\boldsymbol{q_r} \in \mathbb{R}^{N_q}$ denotes the reference minimal coordinates and $\boldsymbol{q} \in \mathbb{R}^{N_q}$ denotes the observed minimal coordinates. Note that the provided system object `sys` has $2N$ bodies but only the first $N$ bodies are actuated. The remaining bodies are passive free joints.

**Definition C.8.** The function `forDyn`(sys, $\boldsymbol{q}, \dot{\boldsymbol{q}}, \boldsymbol{\tau}$) applies forward dynamics in the system `sys` and integrates the minimal coordinates position and velocity vector $\boldsymbol{q}, \dot{\boldsymbol{q}}$ by `sys.T_s`. The function `forKin`(sys, $\mathbf{Q}$) applies forward kinematics in the system `sys`, i.e., it provides a map from minimal $\mathbf{Q}$ to maximal coordinates $\mathbf{T}$. Here, it is additionally vectorized over the time dimension. A text-book reference for both well-known algorithms can be found in Featherstone (2008).

**Definition C.9.** The function `simIMU`($\mathbf{T}, T_s$) simulates a 6D IMU from a trajectory of maximal coordinates. First, the maximal coordinates are butter-worth low-pass-filtered (both the quaternion and position trajectory). Then, a second-order numerical differentiation for both gyroscope and accelerometer is used. The accelerometer is low-pass-filtered. Gravity and simulated noise and bias terms are added. The cutoff frequencies have been optimized such that experimental IMU data is recovered with the highest fidelity from the maximal coordinate trajectories from OMC. Additional details can be found in Bachhuber et al. (2022).

## D RING's Time Complexity and Computational Requirements at Inference

For practical applications, the inference time and computational requirements of RING are critical to enable real-world online applications. Therefore, we conduct a theoretical and empirical time complexity analysis of RING at inference.

The theoretical time complexity of RING depends on the operations involved when advancing the prediction by one timestep. First, we assume a naive matrix multiplication complexity, i.e, let $\mathbf{A} \in \mathbb{R}^{C \times D}$ and $\mathbf{B} \in \mathbb{R}^{D \times E}$ then $\mathbf{AB}$ is $\mathcal{O}(C \times D \times E)$. Now, recall that $N$ is the number of bodies, $F$ the number of features per body (here $F = 9$), $M$ is the message dimension, and $H$ is the hidden state dimension, then

- eq. (4) is $\mathcal{O}(N \times H \times H + N \times 1 \times H^2 + N \times M \times H)$ ($f_\theta$),

- eq. (5) is $\mathcal{O}(N \times M)$ (summation operation, a tree with $N$ nodes has at most $N - 1$ edges),

Table 3: Non-exhaustive list of IMU hardware and their typical (typ.) noise and bias properties as provided in the manufacturers' technical specifications. Unfortunately, not all manufacturers provide this information.

| Hardware | Gyr. N.D. $[°/s/\sqrt{Hz}]$ | Gyr. Of. $[°\,s^{-1}]$ | Acc. N.D. $[\mu g/\sqrt{Hz}]$ | Acc. Of. $[mg]$ |
|---|---|---|---|---|
| Bosch BMI160 (1) | 0.007 | ±3 | 180 | ±40 |
| A.D. ADXL355 (2) | Acc. only | Acc. only | 22.5 | ±25 |
| Xsens MTi 10 (3) | 0.03 | ±0.2 | 60 | ±5 |
| Xsens MTi 100 (3) | 0.01 | ±0.2 | 60 | ±5 |
| Xsens MTw (ours) (4) | 0.01 | N.P. | 200 | N.P. |
| Movella Dot (5) | 0.007 | N.P. | 120 | N.P. |
| K. KXTC9-2050 (6) | Acc. only | Acc. only | 125 | N.P. |
| max (worst-case) | 0.03 | ±3 | 200 | ±40 |

Noise Density (N.D.), Offset (Of.), Not Provided (N.P.)
Accelerometer units are micro-gravity per square-root of Hertz and milli-gravity
Sources in supplementary materials: `bosch_bmi160.pdf` (1), `analog_devices_adxl355.pdf` (2), `xsens_mti.pdf` (3), `xsens_mtw.pdf` (4), `movella_dot.pdf` (5), `kionix_kxtc9-2050.pdf` (6)

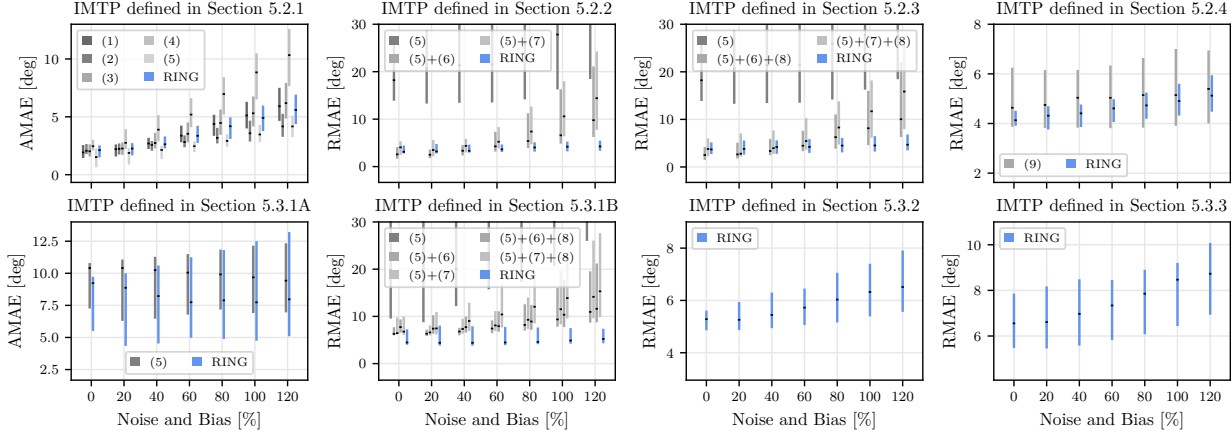

Figure 9: Comparison of the robustness of RING and SOTA methods as the level of noise and bias is increased. To simulate reasonable noise density and bias offset ranges, we constructed a worst-case IMU by combining the worst properties of various IMU manufacturers, as summarized in Table 3. Then, those worst-case noise and bias values are incrementally (in seven equidistant steps) increased from 0 to 120%, and a corresponding amount of simulated noise and bias is added to our real-world IMU data (see Section 5.1), which yields seven modified datasets. RING and all SOTA methods are validated on the modified dataset, and this procedure is repeated using ten different seeds. The AMAEs and RMAEs of all methods are plotted for all IMTPs as a function of the seven steps. The 25%/50%/75%-percentiles across all trials and seeds are shown, and they show that RING maintains accuracy comparable to SOTA methods in all IMTPs and, especially in two-segment KC tracking, substantially outperforms them. Methods are Weber et al. (2021)(1), Madgwick (2010)(2), Mahony et al. (2008)(3), Seel & Ruppin (2017)(4), Laidig & Seel (2023)(5), Laidig et al. (2017)(6), Lehmann (2020)(7),Olsson et al. (2020)(8), Bachhuber et al. (2023)(9).

Table 4: Performance comparison across different hardware configurations.

| Hardware | Computation Time [µs] | | | | Latency [µs] | | | |
|---|---|---|---|---|---|---|---|---|
| | $\lambda_1$ | $\lambda_2$ | $\lambda_3$ | $\lambda_4$ | $\lambda_1$ | $\lambda_2$ | $\lambda_3$ | $\lambda_4$ |
| (1) | $131 \pm 8$ | $757 \pm 143$ | $881 \pm 128$ | $916 \pm 228$ | $158 \pm 7$ | $789 \pm 137$ | $912 \pm 84$ | $975 \pm 172$ |
| (2) | $132 \pm 5$ | $172 \pm 20$ | $181 \pm 17$ | $192 \pm 19$ | $206 \pm 5$ | $241 \pm 25$ | $251 \pm 31$ | $267 \pm 30$ |
| (3) | $78 \pm 5$ | $127 \pm 8$ | $129 \pm 8$ | $127 \pm 9$ | $152 \pm 14$ | $199 \pm 20$ | $197 \pm 11$ | $201 \pm 11$ |
| (4) | $376 \pm 90$ | $681 \pm 148$ | $647 \pm 96$ | $879 \pm 361$ | $603 \pm 167$ | $834 \pm 174$ | $947 \pm 211$ | $1106 \pm 377$ |
| (5) | $214 \pm 55$ | $327 \pm 69$ | $325 \pm 51$ | $338 \pm 60$ | $494 \pm 53$ | $692 \pm 133$ | $684 \pm 426$ | $704 \pm 153$ |

Apple M2 Pro (1), W-1390P (2), W-1390P + RTX A5000 (3), Intel Xeon @ 2.0 Ghz, 1 Core, 2 Threads (Google Colab) (4), Intel Xeon @ 2.0 Ghz, 1 Core, 2 Threads + Tesla T4 GPU (Google Colab) (5)

- eq. (6) is $\mathcal{O}(N \times H \times (2M + F) + N \times H^2)$ (first GRU cell of $g_\theta$), $\mathcal{O}(N \times H)$ (Layernorm), $\mathcal{O}(N \times H \times H + N \times H^2)$ (second GRU cell of $g_\theta$),

- eq. (7) is $\mathcal{O}(N \times H)$ (Layernorm), $\mathcal{O}(N \times H \times H + N \times 1 \times H^2 + N \times 4 \times H)$ ($h_\theta$),

- eq. (8) is $\mathcal{O}(N)$ (normalization).

This leads to an overall complexity of $\mathcal{O}(N \times H \times (H + M + F))$. Note that the leading $N$ term is implemented in a way such that it corresponds to an efficient batch operation and not a for loop. This is crucial for performance (especially on GPUs).

We conduct the empirical analysis for various types of hardware and report computation time and latency required for advancing the prediction by one timestep. Latency includes overheads such as conversion of the NumPy array to the deep-learning-framework-specific array type and potential to-and-from-device transfer overheads. Effectively, latency measures the time required from NumPy array input to NumPy array prediction, i.e.,

$$\underset{\text{np.ndarray}}{\mathbf{X}_t} \overset{\text{to device}}{\to} \underbrace{\underset{\text{jax.Array}}{\mathbf{X}_t} \underbrace{\to \text{ring} \to}_{\text{Computation Time}} \underset{\text{jax.Array}}{\hat{\mathbf{Y}}_t} \overset{\text{to host}}{\to} \underset{\text{np.ndarray}}{\hat{\mathbf{Y}}_t}}_{\text{Latency}}.$$

Consequently, latency needs to be lower than the time delta due to the IMU sampling rate to enable lag-free real-time application (excluding a delay of one frame). Table 4 reports the timings for various hardware and the different IMTPs that include either one-, two-, three-, or four-segment KCs.

# E  Software Example

This example code uses the published software (see Section 4.5) and showcases how RING is applied in Section 5.3.2, i.e., it solves an IMTP that consists of a three-segment KC with sparse 6D IMU attachment and with unknown joint axes directions.

```python
import ring
import numpy as np

T  : int       = 30        # sequence length      [s]
Ts : float     = 0.01      # sampling interval    [s]
B  : int       = 1         # batch size
lam: list[int] = [-1, 0, 1] # parent array; because of Python's conventions body counting starts at 0, as a
    # consequence the base body is indicated by -1 and not 0
N  : int       = len(lam)  # number of bodies
T_i: int       = int(T/Ts) # number of timesteps

X              = np.zeros((B, T_i, N, 9))
# where X is structured as follows:
```

```
13  # X[..., :3]   = acc
14  # X[..., 3:6]  = gyr
15  # X[..., 6:9]  = jointaxis
16
17  # let's assume we have an IMU on each outer segment of the
18  # three-segment kinematic chain
19  X[..., 0, :3]  = acc_segment1
20  X[..., 2, :3]  = acc_segment3
21  X[..., 0, 3:6] = gyr_segment1
22  X[..., 2, 3:6] = gyr_segment3
23
24  ringnet = ring.RING(lam, Ts)
25  yhat, _ = ringnet.apply(X)
26  # yhat: unit quaternions, shape = (B, T_i, N, 4)
```

