# OpenReview forum: "Recurrent Inertial Graph-Based Estimator (RING): A Single Pluripotent Inertial Motion Tracking Solution"
_TMLR — Accepted by TMLR_

### Review · Reviewer_n9Fd · 2024-07-29

**Summary Of Contributions:**

This paper proposes a problem-agnostic ML model for inertial motion tracking (IMT). This is based on representing systems in IMT by a graph and using message passing within an RNN cell to extract hidden representations and making predictions. The parameter sharing within the message passing makes the model independent of the system that it can predict. The network is trained on synthetic data  generated randomly by the Random Chain Motion Generator (RCMG) procedure, yet can achieve zero-shot generalisation to several real-world settings.

**Audience:**

Yes

**Broader Impact Concerns:**

As far as I can see, there are not ethical concerns of the work.

**Claims And Evidence:**

Yes

**Requested Changes:**

- The abbreviation "IMU" is not introduced
- "see 2" (page 4) => "see Figure 2"
- Throughout the paper I find the use of the **Definition** heading a little bit strange. I don't understand how the statements in Definitions 3.2 and 4.1 are definitions. They seem more like remarks.
- Compared to the other figures, I find Figure 2 to be less informative. In particular, the `sys` representation next to the figure doesn't really tell us much. Additionally, the matrix size of __K__ and __$\Gamma$__ is cut off.
- I don't understand the relative-to-parent position array in __Definition 4.2__. In particular, the last sentence. What are the first $N$ values due to the geometry of the KC and what are "sensor-to-body" positions and why do we need it? This can be a little confusing is one is not already familiar with the field.
- Could you please give some more explanation of what the AMAE and RMAE (equations (8) and (9)) tells us? For example, why do we treat  the first component $i=1$ separately from the others and build separate metrics for $i=1$ and $i=2, ..., N$? What is the implication of having a low or high AMAE/RMAE score?
- "This pluripotent behavior origins from a" => "This pluripotent behavior *originates* from a" (page 16).

**Strengths And Weaknesses:**

**Strengths:**
- The introduction of a "pluripotent" model for IMT is an appealing idea and definitely of interest to some of the TMLR audience.
- Experimental results are quite interesting, showing zero-shot generalisation of the general model to concrete real life examples.
- The figures are mostly clear and help to understand the model, which has several moving components.

**Weaknesses:**
- The writing can be hard to understand at certain points. In particular, it requires quite a bit of familiarity within the field of motion tracking, which the general TMLR audience will not have. The paper is most certainly geared towards researchers in robotics than a general ML audience. See also __Requested changes__ below.

---

> ### Author Response · Authors · 2024-08-27
> **Response to Reviewer 1, Part 1**
>
> We thank the reviewer for their valuable feedback.
>
> Reviewer:
> - *The abbreviation "IMU" is not introduced*
> - *"see 2" (page 4) => "see Figure 2"*
> - *"This pluripotent behavior origins from a" => "This pluripotent behavior _originates_ from a" (page 16).*
>
> Response: We thank the reviewer for noticing these potential issues. The acronym IMU is introduced in the beginning of the introduction (*Inertial Measurement Units (IMUs), which typically comprise a 3D accelerometer, a 3D gyroscope, and a 3D magnetometer, have become smaller [...]*). The other two issues are fixed.
>
> ---
> Reviewer:
> - *Throughout the paper I find the use of the **Definition** heading a little bit strange. I don't understand how the statements in Definitions 3.2 and 4.1 are definitions. They seem more like remarks.*
>
> Response: We thank the reviewer for the comment. We agree that both definitions (3.2 and 4.1) can also be interpreted as remarks. Both statements serve to clarify and make specific comments about how that symbol is used in the context of the discussion (this would then also fit the definition of the `sys` object as a collection of several system-related quantities). As such we are open to re-naming the environments to remarks instead of definitions.
>
> ---
> Reviewer:
> - *Compared to the other figures, I find Figure 2 to be less informative. In particular, the `sys` representation next to the figure doesn't really tell us much. Additionally, the matrix size of **K** and **Γ** is cut off.*
>
> Response: We thank the reviewer for noticing this cropping problem. It is fixed.
>
> ---
> Reviewer:
> - *I don't understand the relative-to-parent position array in **Definition 4.2**. In particular, the last sentence. What are the first N values due to the geometry of the KC and what are "sensor-to-body" positions and why do we need it? This can be a little confusing is one is not already familiar with the field.*
>
> Response: We thank the reviewer for the comment. A typical convention for the modelling of rigid-body systems is to specify, for each body, the constant position vector between the zero-points of all DOF in the system (Featherstone, 2008, https://doi.org/10.1007/978-1-4899-7560-7). For all $N$ bodies, these constant position vectors are given by the first $N$ values of the array $\boldsymbol{\text{R}}$. As an example, consider the three-segment KC. The connection vector from hinge joint center one to hinge joint center two is such a constant position vector. But IMUs are typically not attached to the joint center, instead they are attached offset with some position vector in the coordinate system of the body. These position vectors are referred to as the sensor-to-body positions and they are the last $N$ values of the array $\boldsymbol{\text{R}}$. Both set of position vectors are required and must be randomized for optimal sim-to-real transfer. The sensor-to-body position must be randomized to account for an unknown real-world attachment of IMUs. The position vectors that describe the geometry of the chain are required to account for an unknown physical dimensionality of the chain, the, e.g., arm robot can have segments with various lengths. This matters because, following up on the previous example, the vector between the two hinge joints (the length of the middle segment) influences the kinematics of the chain and the measurements of the outer IMUs. We have added a similar clarification to the manuscript:
>
> *[...] a relative-to-parent position array $\boldsymbol{\text{R}}$ that contains the position vector of the body's coordinate system relative to its parent (expressed in the parent's coordinate system). In Section 4.2.1, it is outlined that for each of the $N$ bodies there is a second IMU body. In the $\boldsymbol{\text{R}}$ array, the first $N$ values are used to specify the position vector between two non-IMU bodies (segment-to-segment positions, physical geometry of the KC), while the last $N$ values are used to specify the position vector from non-IMU body to the corresponding IMU body (segment-to-body positions, IMU attachment). This is done to independently randomize both the physical geometry and the IMU attachment, forcing the network to learn to generalize to scenarios such as an arm robot with unknown segment lengths, as well as to calibrate for an unknown IMU attachment. [...]*

---

> ### Author Response · Authors · 2024-08-27
> **Response to Reviewer 1, Part 2**
>
> Reviewer:
> - *Could you please give some more explanation of what the AMAE and RMAE (equations (8) and (9)) tells us? For example, why do we treat the first component i=1 separately from the others and build separate metrics for i=1 and i=2,...,N? What is the implication of having a low or high AMAE/RMAE score?*
>
> Response: We thank the reviewer for the comment. First, it is important to note that magnetometer-free IMT either estimates only the inclination (this is called attitude estimation), or requires exploitation of constraints to correct the heading error in the orientation estimate between two coordinate systems. This is because both accelerometer and gyroscope measurements are invariant under a rotation around the vertical direction (the gravity direction). I.e., if the entire system is rotated by some yaw angle, then all IMU measurements stay the same, and one absolute yaw angle is unobservable.
> Now, because of our choice of $\lambda_N$ in Definition 4.1., the first body ($i=1$) always connects to the base, and the DOF between first body and base is a 6D or free joint. Therefore, magnetometer-free IMT can only be used to estimate the inclination of the orientation between first body and base. For all other bodies ($i>1$), magnetometer-free IMT can exploit constraints in order to correct the heading in the orientation estimate between two bodies.
> This difference is captured by the two metrices AMAE and RMAE. Mathematically, AMAE reports zero angle error if the ground truth orientation and estimated orientation are equal up to an arbitrary heading difference. Whereas, RMAE reports zero angle error if and only if both orientations are exactly identical.
> The implications are that a low AMAE means that the inclination of the system is correctly estimated but the entire system might still be rotated with an arbitrary yaw angle.
> A low RMAE means that the internal pose of the system is precisely estimated, i.e. all orientations between two bodies of the robot are precisely known.
> A high AMAE and RMAE both have the same implication of an incorrect orientation estimate.
> We have added a similar clarification to the manuscript:
>
> *[...] From Section 3.2, recall that magnetometer-free IMT estimates one absolute attitude, and $N-1$ relative orientations.
> This difference is captured by the two metrices AMAE and RMAE. Mathematically, AMAE reports zero angle error if the ground truth orientation and estimated orientation are equal up to an arbitrary heading difference. Whereas, RMAE reports zero angle error if and only if both orientations are exactly identical.
> The implications are that a low AMAE indicates that the inclination of the system is correctly estimated but the entire system might still be rotated with an arbitrary yaw angle.
> A low RMAE means that the entire internal pose of the system is accurately estimated. [...]*

---

### Review · Reviewer_zZGD · 2024-08-09

**Summary Of Contributions:**

The paper introduces the Recurrent Inertial Graph-Based Estimator (RING), a ML-based method for inertial motion tracking (IMT). RING offers a pluripotent, problem-unspecific solution that simplifies the use of IMT technology by eliminating the need for expert knowledge in selecting and parameterizing appropriate methods. The architecture utilizes a decentralized network of message-passing, parameter-sharing RNN that map local IMU data and nearest-neighbor messages to local orientations. This design enables RING to generalize across a broad range of IMT problems and outperforms several state-of-the-art solutions, even under challenging conditions such as magnetometer-free and sparse sensing with unknown sensor-to-segment parameters.

**Audience:**

Yes

**Claims And Evidence:**

Yes

**Requested Changes:**

- My main concern is the robustness of RING in zero-shot learning when dealing with raw IMU data from different sensor configurations. The performance of a model that uses raw IMU data as input can indeed be sensitive to variations in sensor configurations, including differences in calibration, placement, and orientation between devices from different manufacturers. From my experience, the IMU data captured from apple and meta's platforms have large difference. These differences can impact the raw sensor readings, potentially affecting the model's ability to generalize effectively in zero-shot learning scenarios.  The authors either needs to prove their model is robust to sensor configurations and why. Or verify that strategies like domain randomization, data transformation, and fine-tuning may be necessary. [critical to secure my recommendation for acceptance]

- The paper focuses heavily on the accuracy and generalization capabilities of RING but provides limited information on its real-time performance, such as latency, computational efficiency, and resource utilization. These factors are crucial in many practical applications, particularly those requiring real-time feedback or low-latency responses. The authors should evaluate the system for real-time performance.  [critical to secure my recommendation for acceptance]

**Strengths And Weaknesses:**

+ The use of a decentralized network of message-passing RNNs is novel and effectively handles the complexities of various IMT problems. The architecture's ability to generalize across different kinematic chains and sensor setups without retraining is impressive.

+ RING's ability to zero-shot generalize from simulation to experimental data and outperform specialized solutions in various scenarios is a significant contribution.

- Although RING shows remarkable generalization from simulated to real-world data, the reliance on simulated training data may limit its performance in highly specialized or unforeseen real-world scenarios.

- The paper focuses on accuracy but does not provide detailed insights into the real-time performance and computational efficiency of RING, which are critical for practical applications.

---

> ### Author Response · Authors · 2024-08-27
> **Response to Reviewer 2, Part 1**
>
> We thank the second reviewer for their valuable feedback, and for appreciating RING's ability to zero-shot generalize from simulation to experiment.
>
> Reviewer:
> - *My main concern is the robustness of RING in zero-shot learning when dealing with raw IMU data from different sensor configurations. The performance of a model that uses raw IMU data as input can indeed be sensitive to variations in sensor configurations, including differences in calibration, placement, and orientation between devices from different manufacturers. From my experience, the IMU data captured from apple and meta's platforms have large difference. These differences can impact the raw sensor readings, potentially affecting the model's ability to generalize effectively in zero-shot learning scenarios. The authors either needs to prove their model is robust to sensor configurations and why. Or verify that strategies like domain randomization, data transformation, and fine-tuning may be necessary.*
>
> Response:
> We thank the reviewer for the comment. We agree that IMUs in general, and especially across vendors, differ in properties like noise density and bias offset.
> First, it is important to note RNN-based inertial sensor fusion has been demonstrated to generalize across different sensor hardware (Weber et al., 2021, [https://doi.org/10.3390/ai2030028](https://doi.org/10.3390/ai2030028)).
> Nonetheless, to ensure broad real-world applicability, we have added a section to the manuscript where we investigate RING's robustness to different IMU hardware.
> First, we analyze how RING's performance scales as noise and bias properties are incrementally increased. Then, we evaluate RING on openly-available, real-world datasets that use IMUs from different vendors.
>
> We constructed a worst-case IMU by combining the worst properties of various real-world IMU manufacturers, as summarized in Table 3.
> Then, those worst-case noise and bias values are incrementally increased from $0$ to $120\%$, and a corresponding amount of simulated noise and bias is added to our real-world IMU data.
> RING and all SOTA methods are validated on the modified dataset, and this procedure is repeated using ten different seeds.
> Figure 9 [https://github.com/anonymous-sup-material/ring_supplementary_material/blob/main/plot_section_5_5_robustness.pdf](https://github.com/anonymous-sup-material/ring_supplementary_material/blob/main/plot_section_5_5_robustness.pdf) shows the AMAEs and RMAEs of all methods for all IMTPs.
> The figure shows that, whilst the performance of all methods (as expected) slightly worsens as noise and bias are increased, RING maintains accuracy comparable to SOTA methods in all IMTPs and, especially in two-segment KC tracking, substantially outperforms them.
>
> The RepoIMU and OpenAXES Robot Dataset contain real-world IMU and ground truth data.
> Table 2 (see below) reports RING's performance for both external datasets and RING provides consistently low errors demonstrating robustness across different IMU hardware.
>
> As before, all code and data to reproduce these results has been made publicly available in the supplementary material repository.
>
> | Dataset      | IMU                                   | T [s] | Native Rate [Hz] | RMAE [°]    |
> | ------------ | ------------------------------------- | ----- | ---------------- | ----------- |
> | Ours         | Xsens MTw Awinda                      | 402   | 40               | 3.92 ± 1.40 |
> | RepoIMU (1)  | microIMU (2)                          | 390   | 90               | 3.44 ± 0.06 |
> | OpenAXES (3) | Bosch BMI160 + Analog Devices ADXL355 | 227   | ≈ 125            | 2.52 ± 0.29 |
> **Notes:**
> - (1) Szczesna et al., 2016, http://dx.doi.org/10.1007/978-3-319-46418-3_45
> - (2) Jędrasiak et al., 2013, https://doi.org/10.1109/ICIEA.2013.6566403
> - (3) Webering et al., 2023, https://doi.org/_10.25835_/_psqqs9w0_

---

> > ### Author Response · Authors · 2024-08-27
> > **Response to Reviewer 2, Part 2**
> >
> > Reviewer:
> > - *The paper focuses heavily on the accuracy and generalization capabilities of RING but provides limited information on its real-time performance, such as latency, computational efficiency, and resource utilization. These factors are crucial in many practical applications, particularly those requiring real-time feedback or low-latency responses. The authors should evaluate the system for real-time performance.*
> >
> > Response:
> > We thank the reviewer for the comment. We agree that both computational efficiency and latency are critical for real-time applications. To demonstrate RING's real-time capabilities, we have evaluated the computational requirements and latency of RING on various hardware configurations. Specifically, we show that RING enables real-world online application even on low-end hardware. For example, on a single-core Intel Xeon @ 2Ghz, RING can comfortably enable motion tracking of a four-segment KC at more than 500Hz, which is well above typical IMU sampling rates that range from 90Hz to 286Hz (Laidig et al., 2021, https://doi.org/10.3390/data6070072)
> > Table 4 (see below) lists all timings.
> > We have added a paragraph in Section 4.4 that summarises these results. The complete theoretical and empirical time complexity analysis is in Appendix D.
> >
> > | Hardware | Computation Time [μs] λ₁ | C.T. [μs] λ₂ | C.T. [μs] λ₃ | C.T. [μs] λ₄ | Latency [μs] λ₁ | L. [μs] λ₂ | L. [μs] λ₃ | L. [μs] λ₄ |
> > | -------- | ------------------------ | ------------ | ------------ | ------------ | --------------- | ---------- | ---------- | ---------- |
> > | (1)      | 131 ± 8                  | 757 ± 143    | 881 ± 128    | 916 ± 228    | 158 ± 7         | 789 ± 137  | 912 ± 84   | 975 ± 172  |
> > | (2)      | 132 ± 5                  | 172 ± 20     | 181 ± 17     | 192 ± 19     | 206 ± 5         | 241 ± 25   | 251 ± 31   | 267 ± 30   |
> > | (3)      | 78 ± 5                   | 127 ± 8      | 129 ± 8      | 127 ± 9      | 152 ± 14        | 199 ± 20   | 197 ± 11   | 201 ± 11   |
> > | (4)      | 376 ± 90                 | 681 ± 148    | 647 ± 96     | 879 ± 361    | 603 ± 167       | 834 ± 174  | 947 ± 211  | 1106 ± 377 |
> > | (5)      | 214 ± 55                 | 327 ± 69     | 325 ± 51     | 338 ± 60     | 494 ± 53        | 692 ± 133  | 684 ± 426  | 704 ± 153  |
> > **Notes:**
> > - Apple M2 Pro (1)
> > - W-1390P (2)
> > - W-1390P + RTX A5000 (3)
> > - Intel Xeon @ 2.0 GHz, 1 Core, 2 Threads (Google Colab) (4)
> > - Intel Xeon @ 2.0 GHz, 1 Core, 2 Threads + Tesla T4 GPU (Google Colab) (5)

---

### Review · Reviewer_TEny · 2024-08-20

**Summary Of Contributions:**

The paper presents a novel method for Inertial Motion Tracking (IMT) called the Recurrent Inertial Graph-Based Estimator (RING). This method aims to revolutionize the field of IMT by providing a versatile, problem-unspecific solution that does not require expert knowledge for implementation. RING utilizes a decentralized network of message-passing, parameter-sharing recurrent neural networks to process local inertial measurement unit (IMU) data and generate local orientations. The authors claim that RING can effectively tackle a wide range of IMT problems, including those that are magnetometer-free and involve sparse sensing configurations.

Basically, the RING method represents a solid advancement in Inertial Motion Tracking technology, offering a versatile and user-friendly solution that can adapt to a wide range of problems. Its ability to generalize from simulated to experimental data is particularly noteworthy and positions RING as a promising tool for future research and applications. However, further experimental validation would enhance the paper's contributions and facilitate broader adoption. Overall, this work is a solid work.

**Audience:**

Yes

**Broader Impact Concerns:**

No ethical concerns.

**Claims And Evidence:**

Yes

**Requested Changes:**

It is thus suggested to provide time complexity analysis, both theoretically and empirically.

**Strengths And Weaknesses:**

Strengths:

1: The decentralized architecture of RING is a significant advancement in the field of IMT. By allowing for parameter sharing and message passing among networks, RING can adapt to various configurations and problems, showcasing its pluripotency.

2: One of the most impressive aspects of RING is its ability to generalize from simulated training data to real-world experimental data without additional training. This capability is crucial for practical applications, as it reduces the need for extensive data collection and model retraining.

3: The versatility of RING opens up new possibilities for IMT applications across various fields, including biomechanics and autonomous systems. This could lead to wider adoption of motion tracking technologies in previously untapped areas.

Weaknesses:

1: While the paper emphasizes the zero-shot generalization capability, the experimental validation appears to be limited to specific scenarios. More extensive testing across diverse real-world conditions would strengthen the claims made regarding RING's robustness and adaptability.

2: Although RING is designed to be a plug-and-play solution, the underlying architecture may still pose challenges for users without a strong background in machine learning or neural networks. Providing more detailed guidelines or user-friendly tools for implementation could enhance accessibility. It is thus suggested to provide time complexity analysis, both theoretically and empirically.

---

> ### Author Response · Authors · 2024-08-27
> **Response to Reviewer 3**
>
> We thank the third reviewer for their valuable feedback.
>
> Reviewer:
> - *It is thus suggested to provide time complexity analyzis, both theoretically and empirically.*
>
> Response:
> We thank the reviewer for the comment. We agree and have conducted both an empirical and theoretical time complexity analysis of RING. The main findings are summarised in the main manuscript, with the complete analysis in Appendix D.
> In summary, the theoretical time complexity to advance the prediction of RING by one timestep is $\mathcal{O}\left(N \times H \times (H + M + F)\right)$ where $N$ is the number of bodies, $H$ is the hidden state dimension, $M$ is the message dimension, and $F$ is the number of features per body (here $F=9$).
>
> We also conducted an empirical analysis for various types of hardware and report computation time and latency required for advancing the prediction by one timestep.
> Latency includes overheads such as conversion of the NumPy array to the deep-learning-framework-specific array type and potential to-and-from-device transfer overheads. Table 4 (see below) lists all timings.
> In practice this translates to an efficient NN that enables real-world online application even on low-end hardware. For example, on a single-core Intel Xeon @ 2Ghz, RING can comfortably enable motion tracking of a four-segment KC at more than 500Hz, which is well above typical IMU sampling rates that range from 90Hz to 286Hz (Laidig et al., 2021, https://doi.org/10.3390/data6070072)
>
> | Hardware | Computation Time [μs] λ₁ | C.T. [μs] λ₂ | C.T. [μs] λ₃ | C.T. [μs] λ₄ | Latency [μs] λ₁ | L. [μs] λ₂ | L. [μs] λ₃ | L. [μs] λ₄ |
> | -------- | ------------------------ | ------------ | ------------ | ------------ | --------------- | ---------- | ---------- | ---------- |
> | (1)      | 131 ± 8                  | 757 ± 143    | 881 ± 128    | 916 ± 228    | 158 ± 7         | 789 ± 137  | 912 ± 84   | 975 ± 172  |
> | (2)      | 132 ± 5                  | 172 ± 20     | 181 ± 17     | 192 ± 19     | 206 ± 5         | 241 ± 25   | 251 ± 31   | 267 ± 30   |
> | (3)      | 78 ± 5                   | 127 ± 8      | 129 ± 8      | 127 ± 9      | 152 ± 14        | 199 ± 20   | 197 ± 11   | 201 ± 11   |
> | (4)      | 376 ± 90                 | 681 ± 148    | 647 ± 96     | 879 ± 361    | 603 ± 167       | 834 ± 174  | 947 ± 211  | 1106 ± 377 |
> | (5)      | 214 ± 55                 | 327 ± 69     | 325 ± 51     | 338 ± 60     | 494 ± 53        | 692 ± 133  | 684 ± 426  | 704 ± 153  |
> **Notes:**
> - Apple M2 Pro (1)
> - W-1390P (2)
> - W-1390P + RTX A5000 (3)
> - Intel Xeon @ 2.0 GHz, 1 Core, 2 Threads (Google Colab) (4)
> - Intel Xeon @ 2.0 GHz, 1 Core, 2 Threads + Tesla T4 GPU (Google Colab) (5)

---

### Decision · Action_Editor_Kg5n · 2024-10-09

**Recommendation:** Accept with minor revision

**Comment:**

The AE and reviewers support acceptance of this manuscript. The proposed methodology is well-suited for the task and has demonstrated useful generalization properties. Including the additional experiments provided during the author response in a minor revision will further improve the manuscript.

**Audience:**

All reviewers agree there is an audience match for this work in the TMLR community. Additional experiments requested during the response phase provide context for runtime that will likely be of interest for practitioners looking to adopt the proposed method in real-time robot control settings. These and the edits requested by reviewer n9Fd should be added to widen the work's audience.

**Claims And Evidence:**

All reviewers agree the claims made in the manuscript have been clearly supported by empirical evidence -- both in initial reviews and after the author response. The primary contribution to inertial motion tracking is evaluated on diverse kinematic chains and sensor configurations. Performance is appropriately compared with prior work.